# Kinetic fractionation of noble gases in the stratosphere over Japan

**Satoshi Sugawara[1], Ikumi Oyabu[2,3], Kenji Kawamura[2,3], Shigeyuki Ishidoya[4], Shinji Morimoto[5],**

**Shuji Aoki[5], Takakiyo Nakazawa[5], Sakae Toyoda[6], and Hideyuki Honda[5]**

[1]Faculty of Education, Miyagi University of Education, Sendai 980-0845, Japan

[2]National Institute of Polar Research, Tachikawa 190-8518, Japan

[3]Graduate Institute for Advanced Studies, SOKENDAI, Tachikawa 190-8518, Japan

[4]National Institute of Advanced Industrial Science and Technology (AIST), Tsukuba 305-8569, Japan

[5]Center for Atmospheric and Oceanic Studies, Tohoku University, Sendai 980-8578, Japan

[6]School of Materials and Chemical Technology, Institute of Science Tokyo, Yokohama 226-8501, Japan

**Correspondence**: Satoshi Sugawara (sugawara@staff.miyakyo-u.ac.jp)

**Abstract.** Gravitational separation of gas species in the stratosphere is caused mainly by molecular diffusion and is a powerful tool to diagnose stratospheric transport processes. Previous studies have shown that isotopic and elemental ratios of major atmospheric components decrease with increasing altitude in proportion to the differences of their mass numbers. However, there have been no reports of the vertical changes of Kr, Xe, and Ne in the stratosphere. Here we report the results of the first study of the vertical changes of Kr, Xe, and Ne in the stratosphere based on high-precision analyses. Our goal was to reveal the vertical distributions of noble gases and to clarify the mechanisms governing their separations. Noble gases were measured for the stratospheric air collected by balloon-borne cryogenic air samplers over Japan. We found that the isotopic and elemental ratios of all noble gases decreased and increased with increasing altitude for heavy and light noble gases, respectively. Vertical distributions normalized for the mass number differences indicated that the larger the mass number, the smaller the separation of both the isotopic and elemental ratios. The implication was that kinetic fractionation occurred in the stratosphere because of the differences of molecular diffusivities. We performed model simulations and were able to reproduce the kinetic fractionations for heavier noble gases. Results of model simulations suggested that the kinetic fractionations of noble gases were usable as a new tool to diagnose stratospheric transport processes. In the modern atmosphere, it is difficult to detect the long-term change of the stratospheric circulation from noble gases, except for $Ar/N_2$ ratio, in the troposphere. However, it was suggested that changes in the stratospheric circulation during glacial and interglacial cycles may have affected the noble gas elemental ratios in ice core samples.

## 1 Introduction

Earth's atmosphere contains noble gases, which are extremely stable substances. Argon accounts for approximately 0.9 % of the atmosphere by mole fraction and is one of the major constituents of the atmosphere. The other noble gases — He, Ne, Kr, and Xe — exist in the atmosphere, but their mole fractions are very small. Since noble gases are extremely stable

in the atmosphere, their mole fractions can be considered to be almost constant temporally and uniform spatially. $^{40}Ar$ is released from Earth's crust into the atmosphere through radioactive decay of $^{40}K$, but the amount released is extremely small compared to the amount present in the atmosphere (Bender et al., 2008). It can therefore also be considered to have a constant mole fraction in the modern atmosphere. Recent progress in ultra-high-precision analysis has enabled the detection of extremely small variations of the isotopic and elemental ratios of noble gases (Severinghaus et al., 2003; Severinghaus and Battle, 2006; Kawamura et al., 2013; Bereiter et al., 2018a; Oyabu et al., 2025). The possible separation of the noble gases and the major constituents in the atmosphere is generally related to molecular diffusion, which is predominant only in the upper atmosphere above the turbopause and in the air within the firn layer on the surface of polar ice sheets. In both cases, the region is characterized by competition between molecular and eddy diffusion.

In the firn layer, air transport through tortuous open pores is governed mainly by molecular diffusion. The separation of the constituents of the air therefore occurs in proportion to the differences of their mass numbers if the isotopic or elemental ratios in the atmosphere are constant and there are no disturbances caused by eddy diffusion and/or thermal inhomogeneity within the firn. This process is generally called gravitational separation. Under such conditions, the isotopic ratios of $^{29}N_2/^{28}N_2$ and $^{34}O_2/^{32}O_2$ in the firn air increase almost linearly with increasing depth, and the magnitude of separation of $^{34}O_2/^{32}O_2$ is about twice that of $^{29}N_2/^{28}N_2$ (e.g., Schwander et al., 1989; Sowers et al., 1989). At the bottom of the firn layer, air in the open pores is gradually trapped into bubbles in the ice. The component of the air in the ice-core bubbles is altered by this gravitational separation. There are also slight changes because of thermal and eddy diffusion and fractionations that depend on the mass and diameter of the molecule during the bubble formation (Severinghaus and Battle, 2006; Battle et al., 2011). Measurements of various gases in firn air thus provide information that facilitates understanding the process of fractionation during bubble formation and, consequently, the interpretation of the gas component in ice cores.

The processes of advection and eddy diffusion are much more important than molecular diffusion in the troposphere and stratosphere, and it has generally been assumed that molecular diffusion could be ignored at altitudes below about 100 km. The motivation for this study was the discovery in our previous work of the gravitational separation of major atmospheric components in the stratosphere (Ishidoya et al., 2013). Before this discovery, it was commonly assumed that the major atmospheric components and noble gases were uniformly distributed in the troposphere and stratosphere and that their molecular diffusion was an insignificant phenomenon. Their fractionation was assumed to be difficult to detect because of molecular diffusion below the turbopause, except under the special conditions within the lowermost boundary layer (Adachi et al., 2006). Bieri et al. (1970) measured mole fractions of Ne, Ar, and Kr in the upper stratosphere and lower mesosphere by using a rocket-borne cryogenic air sampler and concluded that their mole fractions were identical to those in surface air, that the atmosphere was very well mixed up to the lower mesosphere, and that gravitational separation was too small to be detected. Ehhalt et al. (1975) also measured mole fractions of Ar, Kr, and Xe in the upper stratosphere and showed that their mole fractions were not significantly different in stratospheric and surface air. However, high-precision analytical techniques have recently made it possible to detect even slight separations of major atmospheric components in the stratosphere. The gravitational separation of major atmospheric components in the stratosphere was first reported by Ishidoya et al. (2006). They showed that the variations with altitude of the isotopic and elemental ratios

of the major atmospheric components were caused by differences of molecular mass (Ishidoya et al., 2008a, 2008b, 2013). This explanation was consistent with the mass-dependent relationships among the related molecules such as the ratios of $^{28}N_2/^{29}N_2$, $^{32}O_2/^{34}O_2$, and $^{40}Ar/^{28}N_2$. These mass-dependent fractionations in the stratosphere were similar to those observed in firn air, even though the effects of advection and eddy diffusion far exceed those of molecular diffusion in the stratosphere. Understanding of gravitational separation in the stratosphere has been enhanced by various observations (Ishidoya et al., 2013; 2018; Sugawara et al., 2018) and studies using numerical models (Belikov et al., 2019; Birner et al., 2020). However, the mechanisms and roles of molecular diffusion processes in the stratosphere are not fully understood, mainly because of a lack of relevant data for constituents other than $O_2$, $N_2$, and Ar in the stratosphere.

The discovery of gravitational separation in the stratosphere led us to hypothesize that similar separations would occur in the isotopic and elemental ratios of noble gases in the stratosphere. Actually, measurements of the properties of noble gases in firn air have been conducted and have provided useful information about firn air and bubbles in ice cores (Severinghaus and Battle, 2006; Battle et al., 2011; Kawamura et al., 2013; Buizert et al., 2023; Oyabu et al., 2025). However, with the exception of Ar, there have been no observations of noble gases in the stratosphere. In this study, we report the vertical distribution of noble gases in the stratosphere for the first time, and we discuss the properties associated with molecular diffusion of noble gases in the stratosphere by incorporating the results of numerical models. The main objective of this study was to clarify the mechanisms governing the variation of the properties of noble gases and to understand the processes associated with molecular diffusion in the stratosphere. Understanding the variations of the properties of noble gases in the stratosphere was expected to be greatly facilitated by knowledge derived from studies of firn air. This paper focuses on the similarities and differences between the fractionations in firn air and the stratosphere and discuss disequilibrium fractionation and its role in stratospheric processes.

**2 Experimental Procedure**

**2.1 Sampling stratospheric air**

We have continued to collect samples of stratospheric air over Japan since 1985 using balloon-borne cryogenic samplers (e.g., Nakazawa et al., 1995). In this study, we analyzed air samples obtained from balloon observations during June 2007 and July 2020. Air samplers were launched from the Sanriku Balloon Center (39°10′N, 141°50′E) in Iwate Prefecture in 2007 and the Taiki Aerospace Research Field (42°30'N, 143°26'E) in Hokkaido Prefecture in 2020. Each cryogenic air sampler was equipped with a liquid helium dewar, stainless steel bottles, motor-driven valves, and a control unit (Honda et al., 1996). Liquid helium was used as a refrigerant to enable us to collect stratospheric air cryogenically. We were thus able to collect a large amount of air (20–30 L at standard temperature and pressure) in each bottle. Air samples were collected at 10 different altitudes between 14.5 and 32.8 km during 2007 and at 11 altitudes between 14.8 and 35.4 km during 2020. Approximately two-thirds of the sample air in each bottle was immediately used for measurements of the mole fractions and isotopic ratios of various gases, and one-third was transferred into another stainless-steel container with a volume of 300 mL for long-term archiving and possible use in future studies. We used archived samples of air collected in 2007 for the noble gas measurements. The air samples collected in 2020 were directly aliquoted from each

bottle into two 550-mL Pyrex glass flasks at atmospheric pressure and used for analysis of noble gases. Because analyses
were performed twice for each air sample collected in 2020, the average value was calculated and used for data analysis.
**2.2 Noble gas measurements**
Table 1 summarizes the isotopic and elemental ratios measured in this study. Isotopic and elemental ratios of noble
gases—$\delta(^{40}Ar/^{36}Ar)$, $\delta(^{40}Ar/^{38}Ar)$, $\delta(^{38}Ar/^{36}Ar)$, $\delta(^{86}Kr/^{82}Kr)$, $\delta(^{86}Kr/^{83}Kr)$, $\delta(^{86}Kr/^{84}Kr)$, $\delta(^{132}Xe/^{129}Xe)$, $\delta(^{136}Xe/^{129}Xe)$,
$\delta(^{136}Xe/^{132}Xe)$, $\delta(^{84}Kr/^{40}Ar)$, $\delta(^{132}Xe/^{40}Ar)$, and $\delta(^{22}Ne/^{40}Ar)$—were measured at the National Institute of Polar Research
(NIPR; Tachikawa, Japan). For example, the isotopic ratio of $^{40}Ar$ and $^{36}Ar$ was defined as

$$\delta\left(^{40}_{\square}Ar/^{36}_{\square}Ar\right) = \frac{[n(^{40}Ar)/n(^{36}Ar)]_{sp}}{[n(^{40}Ar)/n(^{36}Ar)]_{rf}} - 1, \tag{1a}$$

where "n", "sp", and "rf" denote the abundance of the respective component, the sample, and the reference gas,
respectively. The isotopic ratios for Ar, Kr, and Xe were defined in a similar manner. The elemental ratio of $^{84}Kr$ to $^{40}Ar$
was defined as

$$\delta\left(^{84}_{\square}Kr/^{40}_{\square}Ar\right) = \frac{[n(^{84}_{\square}Kr)/n(^{40}Ar)]_{sp}}{[n(^{84}_{\square}Kr)/n(^{40}Ar)]_{rf}} - 1. \tag{1b}$$

Because the other elemental ratios were defined similarly, their notations are omitted here. These $\delta$ values are usually
expressed in per meg (1 per meg is 0.001 ‰). For air samples collected in 2007, we measured $\delta(^{40}Ar/^{36}Ar)$, $\delta(^{40}Ar/^{38}Ar)$,
$\delta(^{38}Ar/^{36}Ar)$, $\delta(^{86}Kr/^{82}Kr)$, $\delta(^{86}Kr/^{83}Kr)$, $\delta(^{86}Kr/^{84}Kr)$, $\delta(^{84}Kr/^{40}Ar)$, and $\delta(^{132}Xe/^{40}Ar)$. For air samples collected in 2020,
we also measured $\delta(^{132}Xe/^{129}Xe)$, $\delta(^{136}Xe/^{129}Xe)$, $\delta(^{136}Xe/^{132}Xe)$, and $\delta(^{22}Ne/^{40}Ar)$. Because the analytical method for the
noble gases has been described elsewhere (Severinghaus et al., 2003; Severinghaus and Battle, 2006; Kawamura et al.,
2013; Bereiter et al., 2018a; Oyabu et al., 2025), only a brief description is presented here.
The archived air was stored in a stainless-steel container pressurized to approximately 20 bar and equipped with a
bellows seal valve (Swagelok SS-8BG). An additional bellows seal valve was attached to the existing valve, and an 80-
mL glass flask was connected. After evacuation, an aliquot of the sample was first isolated within the pipette volume
between the bellows seal valves and then expanded into the evacuated 80-mL glass flask. For the samples collected in
2020, a 550-mL glass flask was connected to the 80-mL glass flask. After evacuation, the air sample was expanded into
the 80-mL flask. For the ground-surface values, atmospheric air was sampled outside the NIPR building (hereafter,
"Tachikawa air") in 1500-mL glass flasks using an established method (Oyabu et al., 2020). The 1500-mL flask was
connected to the evacuated 80-mL flask, and the air sample was expanded. For the measurements of samples collected in
2020, Tachikawa air was collected in the same 550-mL glass flasks used for stratospheric samples. The same analytical
procedures were applied to both the stratospheric and Tachikawa air samples to eliminate potential fractionation caused
by sample splitting. No statistically significant differences were observed between the results obtained using the 1500-
mL and 550-mL flasks.
The air sample split into the 80-mL flask was exposed to Zr/Al SAES getters at 900°C for 30–40 min to remove all
the $N_2$, $O_2$, and other reactive gases, followed by an additional 10 min at 300°C to remove $H_2$ gas. The gettered air was
then transferred into a sample tube inserted into a He cycle cooler at temperatures below 10 K for 15 min. After that
transfer, the residual pressure in the vacuum line was measured. For the stratospheric air samples, the residual pressures
were found to be 2.5–20 times those of Tachikawa air. Most of the residual gas consisted of He, which was considered to
be a contaminant introduced during the balloon observation.
The isotopic and elemental ratios of Ar, Kr, Xe, and Ne were measured using a dual-inlet isotope ratio mass
spectrometer (IRMS) (Thermo Fisher Scientific, MAT253). The $\delta(^{132}Xe/^{129}Xe)$, $\delta(^{136}Xe/^{129}Xe)$, and $\delta(^{136}Xe/^{132}Xe)$ ratios
were also measured with the MAT253 using a separate aliquot of the air sample. For the IRMS measurements, the
integration time and idle time were 8 s and 12 s for the Ar isotopes and 26 s and 14 s for the Kr and Xe isotopes. We ran
4 blocks of 16 changeover cycles (sample-standard changeover) for Ar isotopes (64 cycles total), 4 blocks of 25
changeover cycles for Kr isotopes (100 cycles), and 9 blocks of 25 changeover cycles for Xe isotopes (225 cycles). The
cycles were divided into four or nine blocks to enable adjustment of the bellows pressure. Without this adjustment, the
pressure difference between the left and right bellows could increase over time and require a larger correction for the
pressure imbalance. For the $\delta(^{84}Kr/^{40}Ar)$, $\delta(^{132}Xe/^{40}Ar)$, and $\delta(^{22}Ne/^{40}Ar)$ measurements, we used a peak-jumping method
in which the spectrometer magnet setting was sequentially switched between Kr and Ar, Xe and Ar, or Ne and Ar. Each
changeover cycle consisted of a standard and sample measurement at the first magnet setting, followed by the same
sequence at the second setting. The integration time was 8 s for each measurement. We performed 6 cycles for each
isotope ratio, calculated a δ-value for each cycle, and reported the average of these six δ-values as the final value.
The reproducibility of the laboratory measurements was assessed using the pooled standard deviation (SD) of
replicates (measurements made two or more times) for each sample. Table 1 summarizes the pooled SDs of the isotopic
and elemental ratios. Isotopic and elemental ratios reported in this study were measured against reference gases and
normalized against the ground surface values at Tachikawa, Tokyo. Isotopic and elemental ratios of the $N_2$, $O_2$, and Ar—
$\delta(^{29}N_2/^{28}N_2)$, $\delta(^{34}O_2/^{32}O_2)$, $\delta(^{40}Ar/^{36}Ar)$, and $\delta(^{40}Ar/^{28}N_2)$—were also measured at the National Institute of Advanced
Industrial Science and Technology (AIST; Tsukuba, Japan) using IRMS. The technical details of our mass spectrometry
analyses for major atmospheric components have been described by Ishidoya and Murayama (2014). The values of
$\delta(^{29}N_2/^{28}N_2)$, $\delta(^{34}O_2/^{32}O_2)$, and $\delta(^{40}Ar/^{28}N_2)$ are defined as
$$\delta\left(^{29}_{\square}N_2/^{28}_{\square}N_2\right) = \frac{[n(^{29}N_2)/n(^{28}N_2)]_{sp}}{[n(^{29}N_2)/n(^{28}N_2)]_{rf}} - 1, \tag{2a}$$
$$\delta\left(^{34}_{\square}O_2/^{32}_{\square}O_2\right) = \frac{[n(^{34}O_2)/n(^{32}O_2)]_{sp}}{[n(^{34}O_2)/n(^{32}O_2)]_{rf}} - 1, \tag{2b}$$
and
$$\delta\left(^{40}_{\square}Ar/^{28}_{\square}N_2\right) = \frac{[n(^{40}Ar)/n(^{28}N_2)]_{sp}}{[n(^{40}Ar)/n(^{28}N_2)]_{rf}} - 1 \ . \tag{2c}$$
Here, "rf" is the reference gas which is dried natural air filled in a high-pressure cylinder (cylinder no. CRC00045)
(Ishidoya and Murayama, 2014). Table 1 also shows the reproducibility of the $\delta(^{29}N_2/^{28}N_2)$, $\delta(^{34}O_2/^{32}O_2)$, $\delta(^{40}Ar/^{36}Ar)$,
and $\delta(^{40}Ar/^{28}N_2)$ measurements at AIST. The AIST data obtained by balloon observation in 2007 have been
published by Ishidoya et al. (2013).
The Ar isotopic ratios, $\delta(^{40}Ar/^{36}Ar)$, of the air samples obtained by balloon observation and by sampling of firn air at
H128, Dronning Maud Land, East Antarctica were measured independently at NIPR and AIST (Ishidoya et al., 2013;
Oyabu et al., 2025) and compared with each other as shown in Figure 1. The values of $\delta(^{40}Ar/^{36}Ar)$ were negative and
positive in the stratosphere and firn air, respectively, mainly because of the effects of gravitational separations. Because
molecular diffusion is dominant in firn air, the magnitude of fractionation was larger in firn air than in the stratosphere.
The $\delta(^{40}Ar/^{36}Ar)$ values measured by AIST and NIPR were in good agreement in 2020, but there was a systematic
difference between the two measurements in 2007. The mean absolute difference of the values in 2007 was $50 \pm 24$ per
meg. Because the cause of this difference is currently unknown, we used both values in the data analysis.
Some data were significant outliers, probably because of fractionations during sample distribution and/or during the
time the samples were stored in bottles. We therefore fit the data to a linear function of altitude, and we iteratively excluded
outliers if the absolute values of their residuals exceeded $2\sigma$. As a result, 29 data were excluded from a total of 296 data.
**2.3 Mean age of air**
Previous studies have shown that the gravitational separation of the major atmospheric components strengthens with
increasing altitude ($\delta$ values decrease with increasing altitude), and the mean age of stratospheric air increases
simultaneously. Therefore, it is known that the vertical distributions of $\delta(^{29}N_2/^{28}N_2)$ and mean age of air show anti-
correlations (Ishidoya et al., 2013; Sugawara et al., 2018). These correlations have also been reproduced by 3-dimensional
model studies (Belikov et al., 2019; Birner et al., 2020). Furthermore, an anti-correlation in the interannual variations of
the gravitational separation and the mean age of air has been observed in the northern mid-latitude mid-stratosphere
(Ishidoya et al., 2013). This means that gravitational separation becomes stronger when the relevant stratospheric air
becomes older. The mean age of air has been estimated based on observation data of inert trace gases in the stratosphere.
If an inert tracer shows a linear trend in troposphere, the time lag between tropospheric and stratospheric mole fractions
is the mean age of air. The mole fractions of $CO_2$ and $SF_6$ have been widely used for this purpose, because both species
are almost inert in the stratosphere and show monotonous increase trends in troposphere. Therefore, we measured the
mole fractions of $CO_2$ and $SF_6$ in our stratospheric air samples and calculated the mean age of air as described below.
Because this method of estimation has already been described in previous studies (Umezawa et al., 2025; Sugawara et al.,
2025), only a brief description is presented here. The $CO_2$ mole fraction was measured with a nondispersive infrared gas
analyzer at Tohoku University with an analytical precision of less than $0.02~\mu mol~mol^{-1}$. Details about the $CO_2$
measurements have previously been reported (Nakazawa et al., 1995; Aoki et al., 2003; Sugawara et al., 2018). The $SF_6$
mole fraction was measured at Miyagi University of Education (Sendai, Japan) with a gas chromatograph equipped with
an electron capture detector with an analytical precision of less than $0.1~pmol~mol^{-1}$. Details of the $SF_6$ measurements
have been described by Sugawara et al. (2018). The mean age was estimated using the convolution method and the mole

fractions of $CO_2$ and $SF_6$ (e.g., Ray et al., 2014; Fritsch et al., 2020). The convolution method is a method for determining the mean age by calculating the convolution of the age spectrum and a tropospheric reference curve and comparing it with the observed value. Temporal variations of the $CO_2$ or $SF_6$ mole fractions in the stratosphere, $x(\Gamma, t)$, were calculated by convolution of the tropospheric reference curve, $x_0(t)$, and the hypothetical age spectrum, $G(\Gamma, t)$:

$$x(\Gamma, t) = \int_{t-T_B}^{t} x_0(t') G(\Gamma, t-t') dt', \tag{3}$$

where $T_B$ and $G(\Gamma, t)$ are the integration time interval and the age spectrum, respectively. The age spectrum is defined as the statistical probability of individual transit times of different air parcels arrived at a certain place in stratosphere after air parcels intruded into the stratosphere through the tropical upper troposphere. The age spectrum naturally changes over time, but it was ignored here. Because the actual age spectrum is usually unknown, we assumed that it could be approximated by an inverse Gaussian distribution (Waugh and Hall, 2002) as follows:

$$G(\Gamma, t) = \left(\frac{\Gamma^3}{4\pi\Delta^2 t^3}\right)^{1/2} \exp\left[\frac{-\Gamma(t-\Gamma)^2}{4\Delta^2 t}\right], \tag{4}$$

where $\Delta$ denotes the width of the age spectrum. This function is known to be the Green's function of one-dimensional advective diffusion differential equation (Hall and Plumb, 1994). This function will be calculated by assuming the spectral width ($\Delta$) and mean age ($\Gamma$). The mean age is determined by successively calculating the convolutions of equation (3) while varying the value of $\Gamma$ and comparing $x(\Gamma, t)$ with the observed mole fractions. However, $\Delta$ is still unknown. $\Delta$ represents the effect of mixing process in atmospheric transport, and it is expected that the larger the mean age, the greater the effect of mixing and the $\Delta$ will be. Therefore, $\Delta$ is assumed to be given by the relationship $\Delta^2/\Gamma$ = constant (years). This value, called the ratio of moments, is suggested by Hall and Plumb (1994) from the results of a stratospheric AGCM. A ratio of moments of 0.7 years has been widely used in previous studies (e.g. Engel et al. 2002). However, recent progress in studies of age spectra and the ratio of moments has shown that the ratio of moments is currently not well constrained by either models or observations (Garny et al., 2024b). Fritsch et al. (2020) have reported that the mean age calculated from a virtual tracer with a linear trend using a numerical model is in good agreement with observed values when the ratio of moments value of 1.25 is assumed. We assumed the ratio of moments to be 1.25 years, a value reported by Fritsch et al. (2020). The $CO_2$ mole fraction was corrected for $CH_4$ oxidation and gravitational separation prior to the age calculation (Sugawara et al., 2025). The correction of the $CO_2$ mole fraction for gravitational separation is non-negligible and can be estimated by $C \times (m - m_{air}) \times \langle \delta_G \rangle$, where C is $CO_2$ mole fraction, m and $m_{air}$ are the respective mass numbers of the molecule and air, $\langle \delta_G \rangle$ is the average gravitational separation (Ishidoya et al., 2006; Sugawara et al., 2025). The maximum depression of the $CO_2$ mole fraction due to the gravitational separation amounts to about 0.4 $\mu$mol mol$^{-1}$ at the altitudes over 30 km, assuming C = 400 $\mu$mol mol$^{-1}$ and $\langle \delta_G \rangle = -60$ per meg. The tropospheric reference records of $CO_2$ and $SF_6$ mole fractions were prepared by using data from the automatic air sampling equipment used in the Comprehensive Observation Network for Trace gases by Airliner program (Machida et al., 2008; Sawa et al., 2008; Matsueda et al., 2015). Measurements of $CO_2$ mole fraction at altitudes 29 and 31 km in 2020 are not available due to

water contamination into sample air. The overall uncertainties of the mean ages derived from the $CO_2$ and $SF_6$ mole
fractions were estimated to be 0.7 and 0.8 years, respectively (Umezawa et al., 2025).
**3 Results and Discussion**
**3.1 Vertical profiles of noble gases**
Figure 2 shows the vertical profiles of the isotopic and elemental ratios of the noble gases, the $\delta(^{29}N_2/^{28}N_2)$ and
$\delta(^{34}O_2/^{32}O_2)$, and the $CO_2$- and $SF_6$-ages observed in the stratosphere over Japan on 4 June 2007 and 25 July 2020. It
should be noted that the isotopic and elemental ratios of noble gases measured at the NIPR were expressed as the values
relative to the ground surface air at Tachikawa, as described in Section 2.2. However, the $\delta(^{29}N_2/^{28}N_2)$, $\delta(^{34}O_2/^{32}O_2)$,
$\delta(^{40}Ar/^{28}N_2)$, and $\delta(^{40}Ar/^{36}Ar)$ measured at AIST were expressed relative to the values observed in the lowest layer of the
balloon observations. Although there are very few observations of the isotopic and elemental ratios of atmospheric major
components in the upper troposphere, aircraft observations have reported that there is no significant vertical gradient of
$\delta(^{29}N_2/^{28}N_2)$ and $\delta(^{34}O_2/^{32}O_2)$ from the surface to near the tropopause (Ishidoya et al., 2008a). On the other hand, Bent
(2014) observed $\delta(^{40}Ar/^{28}N_2)$ in air samples obtained by the HIAPER Pole-to-Pole Observations (HIPPO) project and
reported a large vertical gradient in troposphere. However, such vertical gradient could not be explained by a 1-D
atmospheric diffusion model, and it was unclear whether it is either natural or artificial. As will be discussed later, our
results of 2-D model also show very small differences between the surface and the upper troposphere. Because the air
samples at the lowest layer were collected below the tropopause in our balloon observations, the differences of the isotopic
and elemental ratios between the ground surface and tropopause should be negligibly small. It is clearly apparent in this
figure that the isotopic and elemental ratios decreased with increasing altitude. The exception was the $\delta(^{22}Ne/^{40}Ar)$ ratio,
which increased with increasing altitude because the difference of the masses of $^{22}Ne$ and $^{40}Ar$ was negative ($-18$ kg
$kmol^{-1}$) for this elemental ratio. Previous studies have reported that $\delta(^{29}N_2/^{28}N_2)$, $\delta(^{34}O_2/^{32}O_2)$, $\delta(^{40}Ar/^{36}Ar)$, and
$\delta(^{40}Ar/^{28}N_2)$ decrease with increasing altitude because of gravitational separation (e.g., Ishidoya et al., 2013). Similar
vertical profiles have also been observed for other isotopic and elemental ratios of noble gases in the stratosphere.
Gravitational separation in the stratosphere, as revealed by observations of isotopic and elemental ratios of major
atmospheric components, indicated that the larger the difference of mass numbers, the greater the separation (Ishidoya et
al., 2013). The isotopic ratios of Ar, Kr, and Xe as well as the elemental ratios of $\delta(^{84}Kr/^{40}Ar)$ and $\delta(^{132}Xe/^{40}Ar)$ observed
in this study showed similar dependencies on mass number differences in the vertical profiles. For example, the mass
number differences of the Kr isotopic ratios—$\delta(^{86}Kr/^{82}Kr)$, $\delta(^{86}Kr/^{83}Kr)$, and $\delta(^{86}Kr/^{84}Kr)$—were 4, 3, and 2 kg $kmol^{-1}$,
respectively, and the observed decreases with altitude were roughly proportional to these values. Because the uncertainties
in the analysis of Xe isotopic ratios —$\delta(^{132}Xe/^{129}Xe)$, $\delta(^{136}Xe/^{129}Xe)$, and $\delta(^{136}Xe/^{132}Xe)$— were much larger compared
to their vertical changes (Fig. 2c), their vertical gradients were not very significant. However, the larger the mass number
difference, the larger the decreasing with altitude, and a mass-dependent relationship similar to that observed for Ar and
Kr isotopic ratios was barely observed. The mass number differences of the elemental ratios $\delta(^{22}Ne/^{40}Ar)$, $\delta(^{40}Ar/^{28}N_2)$,
$\delta(^{84}Kr/^{40}Ar)$, and $\delta(^{132}Xe/^{40}Ar)$ were larger than those of the isotopic ratios, and the magnitudes of fractionation in the
mid-stratosphere were therefore significantly large. Among the elemental ratios, $\delta(^{132}Xe/^{40}Ar)$ had the largest mass
number difference (92 kg kmol$^{-1}$), and the fractionation was about $-2000$ per meg at altitudes above 30 km. In contrast,
the absolute value of the mass number difference for $\delta(^{22}Ne/^{40}Ar)$ was only 18 kg kmol$^{-1}$, which is much smaller than
that of $\delta(^{84}Kr/^{40}Ar)$ or $\delta(^{132}Xe/^{40}Ar)$, but the fractionation was quite large, although the sign of the slope was positive.
As seen in Figure 2 and 3, there are irregular fluctuations of the isotopic and elemental ratios in the vertical
distributions and some of them occur synchronously within the same gas species. Ar isotopic ratios, $\delta(^{40}Ar/^{36}Ar)$,
$\delta(^{40}Ar/^{38}Ar)$ and $\delta(^{38}Ar/^{36}Ar)$ observed in 2007 showed irregularly low values at altitude of 21 km. Kr isotopic ratios
observed also showed similar variations at altitude of 26 km in 2007. Xe isotopic ratios below 21 km also showed similar
variations, although their statistical significance is low. Unfortunately, the cause of these irregular variations is not clear
at present. Because the irregular variations are not common to all gas species, it is likely that there were factors that had
different effects on the gas species. The causes are not necessarily natural, and the possibility of small fractionations
during sample air pretreatments cannot be ruled out. This issue, along with irregular variations in the isotopic and
elemental ratios of the atmospheric major components, remains to be solved in the future.
These results led us to an investigation of the effects of mass number differences. A previous study of the major
atmospheric components has shown that the values normalized by mass number differences (i.e., $\delta(^{29}N_2/^{28}N_2)$,
$\delta(^{34}O_2/^{32}O_2)/2$, $\delta(^{40}Ar/^{36}Ar)/4$, and $\delta(^{40}Ar/^{28}N_2)/12$) show almost the same degree of fractionation (Ishidoya et al., 2013).
The implication is that the separations of these components are almost directly proportional to their mass number
differences. In order to investigate the mass dependencies for noble gases, we defined the value normalized by the mass
number difference, $\delta_n(X/Y)$, as follows:
$$\delta_n\left(\frac{X}{Y}\right) = \frac{\delta(X/Y)}{\Delta m_{X,Y}} \ . \tag{5}$$
Hereafter, X and Y represent molecules associated with the observed ratio. The mass number difference between
molecules X and Y is $\Delta m_{X,Y}$. Figure 3 shows the vertical profiles of $\delta_n(X/Y)$. It is apparent from Fig. 3 that the separations
of $\delta_n(^{29}N_2/^{28}N_2)$ ( $= \delta(^{29}N_2/^{28}N_2)$) and $\delta_n(^{34}O_2/^{32}O_2)$ are large, followed by the isotopic ratios of Ar, Kr, and Xe in decreasing
order. The average values of $\delta_n(X/Y)$ at altitudes above 30 km are summarized in Table 1. The average values of
$\delta_n(^{29}N_2/^{28}N_2)$, $\delta_n(^{34}O_2/^{32}O_2)$, $\delta_n(^{40}Ar/^{28}N_2)$, and $\delta_n(^{40}Ar/^{36}Ar)$ measured by AIST were $-54 \pm 12$, $-52 \pm 9$, $-63 \pm 18$, and
$-48 \pm 6$ per meg, respectively, which are close to the values of gravitational separation reported by Ishidoya et al. (2013).
However, the average values of $\delta_n(^{40}Ar/^{36}Ar)$, $\delta_n(^{40}Ar/^{38}Ar)$, and $\delta_n(^{38}Ar/^{36}Ar)$ measured by the NIPR were $-39 \pm 9$, $-38$
$\pm 9$, and $-40 \pm 11$ per meg, respectively, and those of $\delta_n(^{86}Kr/^{82}Kr)$, $\delta_n(^{86}Kr/^{83}Kr)$, and $\delta_n(^{86}Kr/^{84}Kr)$ were $-23 \pm 1$, $-23 \pm$
4, and $-24 \pm 5$ per meg, respectively. It is apparent from these results that the fractionations normalized by mass number
differences are almost the same for each gas species and that the magnitudes are smaller for Kr than for Ar. Furthermore,
the average values of $\delta_n(^{132}Xe/^{129}Xe)$, $\delta_n(^{136}Xe/^{129}Xe)$, and $\delta_n(^{136}Xe/^{132}Xe)$ were $-16 \pm 4$, $-14 \pm 5$, and $-18 \pm 5$ per meg,
respectively. Their fractionations were even smaller than those of Kr. The average values of the elemental ratios
$\delta_n(^{84}Kr/^{40}Ar)$, $\delta_n(^{132}Xe/^{40}Ar)$, and $\delta_n(^{22}Ne/^{40}Ar)$ were $-27 \pm 4$, $-21 \pm 5$, and $-80 \pm 56$ per meg, respectively. The average

values of $\delta_n(^{84}Kr/^{40}Ar)$ and $\delta_n(^{132}Xe/^{40}Ar)$ were intermediate between those of the isotopes of Kr and Ar and the isotopes of Xe and Ar, respectively. These results indicated that mass-independent fractionations occurred in noble gases and that the deviations from the values for $\delta_n(^{29}N_2/^{28}N_2)$ increased with increasing molecular mass.

One of the noteworthy results was that there were differences between the vertical profiles observed during 2007 and 2020. In particular, the differences were significant for the isotopic ratios of $O_2$, $N_2$, and Ar as well as for the elemental ratios of $\delta(^{40}Ar/^{28}N_2)$, $\delta(^{84}Kr/^{40}Ar)$, and $\delta(^{132}Xe/^{40}Ar)$. In all cases, the fractionations in the mid-stratosphere were greater in 2020 than in 2007. However, no clear difference was apparent in the Kr isotopic ratios. There was also a difference of the $CO_2$ age between 2007 and 2020; the age in 2020 was larger. A previous study of gravitational separations has shown that there is a positive correlation between the $CO_2$ age and the strength of gravitational separations in the mid-stratosphere over Japan (Ishidoya et al., 2013). The implication is that gravitational separations may have been stronger in 2020 than in 2007. However, there was no clear difference of $SF_6$ ages between 2007 and 2020. It has been pointed out that a chemical sink for $SF_6$ in the mesosphere can lead to an overestimation of $SF_6$ age in the mid-stratosphere (e.g., Stiller et al., 2012; Ray et al., 2017; Sugawara et al., 2018; Leedham Elvidge et al., 2018; Garny et al., 2024a). Because the $SF_6$ age estimated in this study was not corrected for the effect of mesospheric loss, it may have been overestimated. Such an overestimation might explain the difference between the $CO_2$ age and $SF_6$ age in 2007. Future observations may lead to a better understanding of the year-to-year variations of noble gases and the mean age of air.

**3.2 Kinetic fractionations**

Studies of fractionations of noble gases have been conducted for firn air on the surface of polar ice sheets and for bubble air in ice core (Severinghaus et al., 2003; Severinghaus and Battle, 2006; Kawamura et al., 2013; Birner et al., 2018; Buizert et al., 2023). Because molecular diffusion is dominant in firn air, unlike the free atmosphere, the deeper the air in the firn, the more fractionations occur due to gravitational separation. Firn air that has reached equilibrium through gravitational separation exhibits mass-dependent fractionation. However, thermal disturbances near the surface cause thermal diffusion, and fluctuations of pressure and wind speed at the ground surface cause eddy diffusion. The result is fractionations that are not governed simply by gravitational separation. In particular, Kawamura et al. (2013) were the first to report that the kinetic fractionations of Kr and Xe isotopic ratios occur by convective mixing in a firn layer because of the competition between eddy and molecular diffusion and the large differences between the molecular diffusivities of Kr and Xe versus those of $N_2$ or Ar. That study was further expanded to consider use of excess-$^{86}Kr$ as a quantitative measure of the degree of gravitational disequilibrium in the firn layer (Severinghaus, 2016; Birner et al., 2018; Buizert and Buizert et al., 2023). This approach in studies of firn air is expected to be of great help in interpreting the fractionations of noble gases in the stratosphere discovered in the present study.

Advective flow and eddy diffusion are predominant in the atmosphere, unlike firn air, and therefore stratospheric air is essentially in a state of gravitational disequilibrium with respect to molecular diffusions. The implications of this fact can be applied to the gravitational separation of the major components of the atmosphere, which is quite different from the fractionations determined by the gravitational equilibrium in firn air. Fractionation in gravitational equilibrium ($\delta_{GE}$)

is given by Eq. (6):
$$\delta_{GE} = \exp\left(-\frac{\Delta mgz}{RT}\right) - 1,$$ (6)
where $\Delta m$, g, R, and T denote the mass number difference, gravitational acceleration, the gas constant, and temperature
(K), respectively. The variable z is the altitude (m) above the ground, but it can also be the depth in a firn if it is negative.
Because the lowermost z value is about $-100$ m for firn air, $\frac{\Delta mgz}{RT}$ is less than $10^{-3}$. Equation (6) can therefore be
approximated by Eq. (7):
$$\delta_{GE} \approx -\frac{\Delta mgz}{RT}.$$ (7)
It is clear from Eq. (7) that gravitational equilibrium is dependent on mass in firn air. For example, the value of $\delta_{GE}$ is
about $-4.7$ per meg m$^{-1}$ at $\Delta m = 1$ and T = 250 K. This value is typical of gravitational separations in a firn and is
equivalent to 0.47 per mil enrichment at z = $-100$ m relative to the surface of a firn. In the stratosphere, typical
gravitational separation at an altitude of about 35 km in the mid-latitudes, which is approximately 20 km higher than the
height of the tropopause, is about $-60$ per meg (Ishidoya et al., 2013; Sugawara et al., 2018). The vertical gradient for this
fractionation is about $-0.003$ per meg m$^{-1}$. The magnitude of gravitational separation in the stratosphere is therefore
roughly 1/1900–1/1700 of the degree of gravitational equilibrium at temperatures of 210–230 K. The implication is that
the major difference between firn air and the stratosphere is the degree of gravitational disequilibrium. In this almost
overwhelming state of disequilibrium, $N_2$, $O_2$, and Ar are undergoing mass-dependent fractionations in the stratosphere,
probably because the diffusivities of these molecules are roughly the same. However, the diffusivities of Ne, Kr, and Xe
are significantly different from those of the major components of the atmosphere, and kinetic fractionation is therefore
thought to be prominent in the stratosphere.

353        In previous studies of the isotopic and elemental ratios of the major components of the atmosphere, $\langle\delta_G\rangle$ has been

defined to be an average gravitational separation normalized by the difference of mass numbers and has been
evaluated as follows:
$$\langle\delta_G\rangle = \frac{1}{3}\left[\delta_n\left(\frac{^{29}N_2}{^{28}N_2}\right) + \delta_n\left(\frac{^{34}O_2}{^{32}O_2}\right) + \delta_n\left(\frac{^{40}Ar}{^{28}N_2}\right)\right].$$ (8)
In Eq. (8), $\delta$ values are always taken to be zero in the troposphere and are expressed as deviations from zero. Equation
(8) is based on the observation that the decreases of the three isotopic and elemental ratios with altitude are dominated by
mass-dependent processes. It has also been confirmed by numerical models that gravitational separations form a mass-
dependent vertical profile for these three ratios (Ishidoya et al., 2013). The gravitational separation can be expressed
generally using $\langle\delta_G\rangle$ as follows:
$$\delta_G\left(X/Y\right) = \Delta m_{X,Y} \times \langle\delta_G\rangle.$$ (9)
However, $\delta(X/Y)$ values for the heavy noble gases are clearly smaller than the mass-dependent value of $\delta_G(X/Y)$, as

described above. We considered the similarity of fractionation to the gravitational disequilibrium in firn air and introduced

a quantity that represented the magnitude of the kinetic fractionations in the stratosphere. The values of $r(X/Y)$ and

$\psi(X/Y)$ corresponding to the isotopic or elemental ratio of $\delta(X/Y)$ were defined as follows:

$$r\left(X/Y\right) = \frac{\delta_n(X/Y)}{\delta_n\left(\frac{^{29}N_2}{^{28}N_2}\right)}, \tag{10}$$

$$\psi\left(X/Y\right) = r\left(X/Y\right) - 1. \tag{11}$$

The value of $r(X/Y)$ is the ratio of $\delta_n(X/Y)$ relative to $\delta_n(^{29}N_2/^{28}N_2)$. If the variation of $\delta(X/Y)$ were due to only

gravitational separation, then $r(X/Y)$ and $\psi(X/Y)$ would be expected to equal 1 and zero, respectively. Figure 4 shows the

relationship between $\delta_n(X/Y)$ and $\delta_n(^{29}N_2/^{28}N_2)$. The values of $r(X/Y)$ were determined by fitting a linear function to the

relationship between the two values. Values of $\psi(X/Y)$ are quantities that generalize the concept of excess-$^{86}$Kr ($^{86}$Kr$_{XS15}$)

used in the ice core analysis by Buizert et al. (2023) to all other isotopic and elemental ratios. For the heavy noble gases,

the process of air mixing due to eddy diffusion produces gravitational disequilibrium, and the values of $\psi(X/Y)$ are always

negative. Table 1 summarizes the values of $r(X/Y)$ and $\psi(X/Y)$. The values of $r(X/Y)$ were roughly close to 1.0 for the

$\delta(^{34}O_2/^{32}O_2)$, $\delta(^{40}Ar/^{28}N_2)$, and Ar isotopic ratios, but they were clearly smaller than 1.0 for the heavy noble gases. For

example, if we compare $\delta(^{40}Ar/^{36}Ar)$, $\delta(^{86}Kr/^{82}Kr)$, and $\delta(^{136}Xe/^{132}Xe)$, which all have mass number differences of 4 kg

kmol$^{-1}$, the heavier the noble gas, the lower $r(X/Y)$. As shown in Table 1, the $r(X/Y)$ values decreased significantly in

the order Ar, Kr, and Xe; the lowest values were 0.24–0.38 per meg (per meg)$^{-1}$ for the Xe isotopic ratios. These facts

suggested that the kinetic fractionations are not all the same and depend on the element.

If the anomalously low $r(X/Y)$ and $\psi(X/Y)$ values of the heavy noble gases were due to kinetic fractionations,

differences of molecular diffusivities would likely be important. We therefore compared the diffusivities of all molecules

involved in this study and investigated the relationships between the observed fractionations and diffusivities. According

to Reid et al. (1987), the diffusivity of molecule X in air is given by Eq. (12):

$$D_{X,air} = 1.43 \times 10^{-4} \frac{T^{1.75}}{p} \frac{1}{\phi_{X,air}} \quad (\text{m}^2 \, \text{s}^{-1}), \tag{12}$$

where $T$ and $p$ denote temperature (K) and pressure (hPa), respectively. The variable $\phi_{X,air}$ is independent of temperature

and pressure and is given by Eq. (13):

$$\phi_{X,air} = \sqrt{m_{X,air}}\left(\sqrt[3]{\sigma_X} + \sqrt[3]{\sigma_{air}}\right)^2, \tag{13}$$

where $\sigma_X$ and $\sigma_{air}$ are the diffusion volumes of a molecule X and of air, respectively. The diffusion volume parameter was

originally introduced by Fuller et al. (1966, 1969) for a semi-empirical method to estimate diffusivity. We used values of

atomic diffusion volume parameters listed in Table 11-1 of Reid et al. (1987). The variable $m_{X,air}$ is the harmonic mean of

mass numbers for molecule X and air and is given by Eq. (14):

$$m_{X,air} = 2\left(\frac{1}{m_X} + \frac{1}{m_{air}}\right)^{-1}. \tag{14}$$
Table 2 summarizes these values along with the ratios of diffusivities relative to $^{28}N_2$. In Table 2, the value of $D_{X,air}$ is an
example at 1000 hPa and 273 K, but the ratio $D_{X,air}/D_{28N2,air}$ is independent of temperature and pressure. The lightest and
heaviest molecules, $^{22}Ne$ and $^{136}Xe$, have the largest and smallest diffusivity ratios of 1.495 and 0.634, respectively. The
diffusivity therefore varies greatly as a function of the mass of the molecule, and that dependence is an important cause
of kinetic fractionation. Although molecular diffusivities are dependent on molecular masses, the relationship between
them is not approximated by a simple function. Furthermore, the value of $\delta(X/Y)$ is influenced by the individual molecular
diffusivities of X and Y in air. To understand the relationship between kinetic fractionations and molecular diffusivities,
the diffusivities of both X and Y must therefore be considered. As shown in Table 1, the values of $\psi(^{84}Kr/^{40}Ar)$ and
$\psi(^{132}Xe/^{40}Ar)$ are intermediate between the values of $\psi$ for Ar and Kr and for Ar and Xe, respectively. We therefore
transformed Eq. (13) and introduced $\phi_{X,Y}$ corresponding to $\delta(X/Y)$ as follows:
$$\phi_{X,Y} = \sqrt{m_{X,Y}}(\sqrt[3]{\sigma_X} + \sqrt[3]{\sigma_Y})^2, \tag{15}$$
where $m_{X,Y}$ is the harmonic mean of the mass numbers for molecules X and Y. The diffusivity factor, $\mu_{X,Y}$, is then defined
as the ratio of $\phi_{29N2,28N2}$ to $\phi_{X,Y}$ as follows:
$$\mu_{X,Y} = \frac{\phi_{29N_2, 28N_2}}{\phi_{X,Y}}. \tag{16}$$
This diffusivity factor equals the ratio $D_{X,Y}/D_{29N2,28N2}$, which is independent of temperature and pressure. Table 1 shows
the values of $m_{X,Y}$ and $\mu_{X,Y}$. Figure 5a shows the correlation between $\mu_{X,Y}$ and the observed $r(X/Y)$. It is apparent from
this figure that $r(X/Y)$ is roughly proportional to $\mu_{X,Y}$. The implication is that the kinetic fractionations of noble gases in
the stratosphere can be explained by the differences of their molecular diffusivities.

412        Figure 6 is a schematic representation that explains the vertical distribution of the gravitational separations and kinetic

fractionations in firn air and the stratosphere. As mentioned above, the results from firn air have been extremely useful in
interpreting the kinetic fractionation of the stratosphere, and the differences between firn air and the stratosphere have
also been clarified by this study. In firn air, the environment is basically close to gravitational equilibrium, and the
magnitude of kinetic fractionation is relatively small. As shown in Fig. 6, the value of $\delta(^{29}N_2/^{28}N_2)$ at the bottom of the
firn layer varies as a function of the thickness of the diffusive zone, but the typical value is ~0.4 ‰ (e.g., Landais et al.,
2006). In contrast, because the zone in which kinetic fractionation occurs is a narrow layer between the well-mixed and
diffusive zones, the difference between $\delta_n(X/Y)$ and $\delta(^{29}N_2/^{28}N_2)$ is roughly −20 and −30 per meg for the Kr and Xe
isotopic ratios, respectively, at the bottom of a firn (Kawamura et al., 2013). If we calculate the excess value from these
values, then (for example) $\psi(^{86}Kr/^{82}Kr)$ is approximately −50 per meg ‰$^{-1}$. Buizert et al. (2023) have reported a
$\psi(^{86}Kr/^{82}Kr)$ value for the last 25 kilo-years from an analysis of an ice core from the West Antarctic Ice Sheet Divide and
have used it as a proxy for dispersive mixing due to barometric pumping in firn air. They have reported $\psi(^{86}Kr/^{82}Kr)$
values that range from 0 to −60 per meg ‰⁻¹. However, the fractionations in the stratosphere observed in this study were
−60 and −25 per meg for $\delta_n(^{29}N_2/^{28}N_2)$ and $\delta_n(^{86}Kr/^{82}Kr)$, respectively, at an altitude of ~35 km (Fig. 3). The difference,
35 per meg, is roughly close to the value at the bottom of the firn, but it is comparable to the fractionation of $\delta_n(^{29}N_2/^{28}N_2)$
in the stratosphere. Therefore, $\psi(^{86}Kr/^{82}Kr)$ is about 0.6 per meg (per meg)⁻¹ (= 600 per meg ‰⁻¹). The implication is that
the excess value is roughly tenfold those in firn air. The explanation is that, unlike firn air, the atmosphere is in a state of
significant disequilibrium throughout all of its layers, and gravitational separation and kinetic fractionation have
comparable effects on the fractionation of heavy noble gases.
As shown in Fig. 5a, the relationship between $r(X/Y)$ and $\mu_{X,Y}$ in some cases is nonlinear. The elemental ratios and
the isotopic ratios of Ar tend to have smaller $r(X/Y)$ values than those predicted from a linear relationship between $r(X/Y)$
and $\mu_{X,Y}$. The reason for the nonlinearity is unclear at this time, but one of the possible reasons is an effect of thermal
diffusion. The effects of thermal diffusion in the stratosphere have not been studied and are not well understood for noble
gases. It is presently considered difficult to clearly quantify the contribution of fractionation due to thermal diffusion in
the stratosphere.

### 3.3 Two-dimensional model

We hypothesized that the vertical profiles of noble gases revealed in this study reflected the effects of not only
gravitational separation but also kinetic fractionation due to differences of molecular diffusivities, as described above. To
verify this hypothesis, we performed numerical simulations using a two-dimensional model of the middle atmosphere
(SOCRATES) developed by the National Center for Atmospheric Research (Huang et al., 1998; Park et al., 1999;
Khosravi et al., 2002). This model has previously been used to reproduce the gravitational separation of the atmospheric
major components (Ishidoya et al., 2013; Sugawara et al., 2018). Only a brief description of the method is therefore given
here. To reproduce fractionations of stratospheric noble gases, the flux associated with molecular diffusion must be
calculated. The vertical component of the flux due to molecular diffusion for molecular species X is given by
$$F_{X,z} = -D_{X,air}\left\{\frac{\partial n_X}{\partial z} + \frac{m_X g}{RT}n_X + (1+\alpha_{TX})\frac{\partial(\ln T)}{\partial z}n_X\right\}, \tag{17}$$
where $n_X$, $m_X$, and $\alpha_{TX}$ are the number density, molecular mass, and thermal diffusion factor of species X, respectively,
and g, R, and T denote the acceleration of gravity, the gas constant, and temperature, respectively (Banks and Kockarts,
1973). As described before, the process of thermal diffusion in the stratosphere is presently unclear and was ignored in
this study. Because noble gases were not included in the original SOCRATES model, we updated it to perform calculations
for all molecules shown in Table 2. The mole fraction was calculated for each molecule, and then the isotopic and
elemental ratios $\delta(X/Y)$ were calculated offline. Equation (12) was used to calculate the molecular diffusivity, $D_{X,air}$. All
$\delta(X/Y)$ at the ground surface were set to zero as a boundary condition. However, this boundary condition was not used
when we considered tropospheric enrichment (see Section 3.4). In addition to the noble gases, we added a virtual clock
tracer to the model. The tracer increases in proportion to elapsed time at the ground surface and was used to calculate the
mean age of air. A previous study has shown that the intensity of Brewer–Dobson circulation in SOCRATES is too large
to reproduce the mean age of stratospheric air (Sugawara et al., 2018). We therefore arbitrarily reduced the mass stream
function of the residual mean meridional circulation (hereafter, RMC) to calculate a realistic mean age. This change also
improved the reproduction of gravitational separation. We used a 40-year spin-up calculation to achieve steady state
fractionations and then ran the simulation for another 20 years ("control run"). The simulation time step was 5 days. The
monthly average values in the last 5 years were compared with observations. Although the observations were carried out
in different months (i.e., June 2007 and July 2020), the differences of the monthly averages simulated for June and July
were small. We therefore simply averaged the $\delta$ values simulated for those months.
The average meridional distributions simulated by SOCRATES are shown in Fig. 7 for $\delta(^{29}N_2/^{28}N_2)$, $\delta(^{40}Ar/^{36}Ar)$,
$\delta(^{86}Kr/^{82}Kr)$, and $\delta(^{132}Xe/^{129}Xe)$ and in Fig. 8 for $\delta(^{84}Kr/^{40}Ar)$, $\delta(^{132}Xe/^{40}Ar)$, and $\delta(^{22}Ne/^{40}Ar)$ together with the results
for the mean age of air (Fig. 8d). The simulated vertical profiles at 40°N were shown in Figure 2 and 3. Figure 4 shows
the relationships between the $\delta_n(X/Y)$ values and $\delta_n(^{29}N_2/^{28}N_2)$ simulated at 40°N. It is apparent from these figures that
the simulated fractionations closely reproduced the observed results in 2007, but the simulations slightly underestimated
the observed results in 2020. The most important point was that the model simulations basically reproduced smaller and
larger $\delta_n(X/Y)$ values of the heavy (Kr and Xe) and light (Ne) noble gases, respectively, compared with the $\delta_n(^{29}N_2/^{28}N_2)$
(Fig. 4). Both the mass-dependent gravitational separations and the kinetic fractionations were therefore reproduced by
the model. Figure 5b shows the relationship between $r(X/Y)$ and the diffusivity factor $\mu_{X,Y}$ reproduced by the model.  It
is apparent in this figure that the values of $r(X/Y)$ simulated for the isotopic and elemental ratios of major atmospheric
components, $\delta(^{84}Kr/^{40}Ar)$ and $\delta(^{132}Xe/^{40}Ar)$, were related almost linearly to $\mu_{X,Y}$. However, the fact that the values of
$r(X/Y)$ simulated for the isotopic ratios of Kr and Xe were closer to 1.0 than the observed results meant that the kinetic
fractionations were underestimated for the isotopic ratios of heavy molecules. The reason for these underestimations of
kinetic fractionation is presently unclear. Only the mean circulation was arbitrarily reduced in these simulations, and the
eddy diffusion coefficient was not changed. Such adjustments may have been unrealistic and contributed to the
underestimation of kinetic fractionation. The effects of eddy diffusions on kinetic fractionations are discussed in Section

480 3.5.

To investigate the kinetic fractionation, the relative contributions of molecular and eddy diffusion are important. In an
environment dominated by molecular diffusion alone, gravitational separation according to the mass number difference
of molecules occurs, whereas in an environment dominated by eddy diffusion alone, separation according to molecular
species is not expected to occur. The Péclet number (Pe) is a dimensionless number defined as the ratio of the transport
rates of advection and diffusion. It is known that kinetic fractionations occur in firn air under conditions where molecular
and eddy diffusivities are in competition (Kawamura et al., 2013) and the Pe is approximately 1. Such conditions in
Earth's atmosphere appear only in the upper atmosphere at altitudes greater than 100 km. It is therefore likely that large
kinetic fractionations occur in the upper atmosphere at altitudes above 100 km. However, the predominance of advection
and eddy diffusion in the troposphere and stratosphere means that the Pe must be much larger than 1. As is apparent from
Eq. (17), the flux associated with molecular diffusion in the atmosphere is usually considered only in the vertical direction.
However, three-dimensional advection and eddy diffusion are essential in atmospheric transport processes. It is therefore
difficult to define Pe in the free atmosphere. Given that fractionations occur mainly in the vertical direction and that the
similarity with the Pe in firn air (Buizert et al., 2023) was considered, we approximated the Pe of molecular species X in
the atmosphere as follows:
$Pe_X = \frac{|w|H + K_{zz}}{D_X}$,                                                                                                        (18)
where w, H, and $K_{zz}$ denote the vertical component of the RMC, atmospheric scale height (=RT/$m_{air}$g), and vertical eddy
diffusion coefficient, respectively. We decomposed $Pe_X$ into its advection and eddy diffusion components ($Pe_{X,w} = \frac{|w|H}{D_X}$
and $Pe_{X,K} = \frac{K_{zz}}{D_X}$) and simulated them for $^{28}N_2$, as shown in Fig. 9. Because atmospheric pressure decreases exponentially
with increasing altitude and molecular diffusivity is inversely proportional to atmospheric pressure, both components of
the Pe values are O($10^2$)–O($10^3$) in the mid-stratosphere. They decrease with increasing altitude and are O($10^{-1}$)–O($10^0$)
at altitudes of ~100 km. The high Pe numbers in the mid-stratosphere compared to firn air seem to contradict the fact that
kinetic fractionations are observable in the stratosphere. This apparent contradiction may be related to the difference in
the vertical ranges of both observations: the range is about O($10^1$–$10^2$) m in the firn but about O($10^4$) m in the stratosphere.
Small kinetic fractionations may therefore be observed at high altitudes up to about 35 km. These results imply that
observing kinetic fractionation in the lower stratosphere may be difficult because the Pe is large, and the vertical range is
narrow.

507        Although the Pe is usually defined as a positive value, the actual vertical component of the RMC is upwelling (w >

0) inside the low-latitude "tropical pipe" (e.g., Neu and Plumb, 1999) and downwelling (w < 0) at higher latitudes. The
relationship between $Pe_{X,w}$ and kinetic fractionation is therefore expected to differ between the tropics and higher latitudes.
In this relationship, the magnitude of gravitational separation in the stratosphere is much smaller above the equatorial
region than in the mid-latitudes (Sugawara et al., 2018). Unfortunately, there are no observations of noble gases above
the equator, but our model calculations shown in Figs. 7 and 8 predict that the $\psi$(X/Y) values of heavy noble gases are
slightly lower in the equatorial mid-stratosphere than in the mid-latitude mid-stratosphere. The implication is that kinetic
fractionations vary with latitude in the stratosphere.

515        Although our model results tended to underestimate the kinetic fractionations of heavy noble gases, the observed

results that their fractionations were smaller than those expected from mass-dependent gravitational separation were
reproduced well in our simulation, in which large differences of molecular diffusivities were essential for the kinetic
fractionation of noble gases in the stratosphere. In the following sections, we use the model to further extend our
investigations to possible tropospheric enrichments driven by stratospheric fractionations (Section 3.4) and to the
sensitivity of the fractionation of noble gases to stratospheric transport processes (Section 3.5).
**3.4 Tropospheric enrichments and possible variations of tropospheric noble gases**
The isotopic and elemental ratios of noble gases in the troposphere were set to zero in this study because we expected
that their variations in the troposphere would be negligibly small. However, if we assume that the total amount of noble
gases in the atmosphere is conserved, the fact that the $\delta$(X/Y) of noble gases decreases monotonically with altitude
(increases for $^{22}Ne$) due to fractionation in the stratosphere suggests that the noble gases are enriched (diluted for $^{22}Ne$)
at the ground surface. In addition, a change over a long period of time of atmospheric transport processes such as the
Brewer–Dobson circulation in the stratosphere would change the fractionations of noble gases, not only in the stratosphere
but also in the troposphere. This idea has been used to interpret long-term variations of $\delta(^{40}Ar/^{28}N_2)$ in the troposphere.
That interpretation has shown that changes of the Brewer–Dobson circulation can affect the long-term trend of
$\delta(^{40}Ar/^{28}N_2)$ near the ground surface (Ishidoya et al., 2021). In this study, the same approach was applied to the isotopic
and elemental ratios of all observed noble gases, and model simulations were performed for several scenarios. We
expressed the total amount of molecular species X in the atmosphere as N(X). By replacing the reference values in Eqs.
(1) and (2) with the ratio of N values, we newly defined $\delta_\Omega(X/Y)$ as follows:
$$\delta_\Omega(X/Y) = \frac{[n(X)/n(Y)]_{sp}}{[N(X)/N(Y)]_{\square}} - 1. \tag{19}$$

If $\Delta m_{X,Y} > 0$, this value will be positive near the ground surface and negative at altitudes above the lower stratosphere. It
is difficult to obtain the value of $\delta_\Omega(X/Y)$ from atmospheric observations, but it can be calculated using numerical models.
For example, Fig. 10 shows the values of $\delta_\Omega(^{132}Xe/^{40}Ar)$ calculated by SOCRATES. It is apparent from this figure that
the enrichment near the surface is 132–138 per meg and that an isosurface where $\delta_\Omega(^{132}Xe/^{40}Ar)$ equals zero exists near
the lower stratosphere. It is also apparent that the altitude of the zero isosurface is higher near the equator and lower at
high latitudes. This latitudinal dependence is due to Brewer–Dobson circulation. Because tropospheric air is well mixed,
there is little dependence of $\delta_\Omega(^{132}Xe/^{40}Ar)$ on altitude within the troposphere, but it decreases rapidly above the
tropopause. The fact that the latitudinal difference at the ground surface is less than 8 per meg implies that it is difficult
to detect latitudinal differences with the current observational precision. Figure 11 shows the vertical profiles of $\delta_\Omega(X/Y)$
for various ratios at 40°N. Table 3 shows the annual mean values of $\delta_\Omega(X/Y)$ at the ground surface in the control run. The
values of $\delta_\Omega(^{29}N_2/^{28}N_2)$ and $\delta_\Omega(^{40}Ar/^{28}N_2)$ at the ground surface were 2.4 and 28 per meg, respectively. The value
normalized by the mass number difference for $\delta_\Omega(^{40}Ar/^{28}N_2)$ was therefore 28/12 = 2.3 per meg. The enrichment was
therefore almost directly proportional to the mass difference. In contrast, the normalized value of $\delta_\Omega(^{132}Xe/^{40}Ar)$ at the
ground surface was 1.44 per meg, which was clearly smaller than the $\delta_\Omega(^{29}N_2/^{28}N_2)$ of 2.4 per mg. This difference was
due to the effect of kinetic fractionation in the upper air, and the corresponding $\psi(^{132}Xe/^{40}Ar)$ value was −0.40 per meg
(per meg)$^{-1}$. These results suggested that kinetic fractionation in the stratosphere influenced tropospheric fractionation,
although its magnitude was much smaller in the troposphere than in the stratosphere.

552         We carried out additional simulations to investigate the possibility that the value of $\delta_\Omega(X/Y)$ at the ground surface

changed with the change of stratospheric circulation. If the Brewer–Dobson circulation strengthened with time, the
stratosphere–troposphere exchange would also be enhanced, and that enhancement would act to homogenize $\delta_\Omega(X/Y)$
and simultaneously reduce the mean age of stratospheric air. Conversely, if the stratospheric circulation weakened, the
vertical differences of $\delta_\Omega(X/Y)$ and the mean age would increase. We therefore performed model simulations in which
we changed the RMC in SOCRATES using a method similar to the method applied by Ishidoya et al. (2021). Although
many studies have been conducted on the mean age of stratospheric air and possible changes of transport processes,

whether the mean age in the mid-stratosphere is increasing or decreasing is unclear at present (e.g., Engel et al., 2009; Garny et al, 2024b). For example, Diallo et al. (2012) have reported that the mean age was increasing at a rate of about 0.3 years decade$^{-1}$ in the mid-stratosphere during the period 1989–2010 based on the ERA-Interim reanalysis data. However, Engel et al. (2017) have reported that there was no significant trend of the age of air observed by high-altitude balloons during the period 1975–2016; the age increased very slightly at a rate of $+0.15 \pm 0.18$ years decade$^{-1}$, which was supported by more recent result of $CO_2$-age observation (Sugawara et al, 2025). Based on these results, the RMC in the model was gradually weakened during the 20-year period so that the mean age of air increased by 0.15 years decade$^{-1}$ at an altitude of 35 km over the northern mid-latitudes ("weakened-RMC" scenario) to evaluate the sensitivities of the noble gases to this change. The results are shown in Fig. 12a for the mean age and in Fig. 12b–f for $\delta_\Omega(^{29}N_2/^{28}N_2)$, $\delta_\Omega(^{40}Ar/^{28}N_2)$, $\delta_\Omega(^{84}Kr/^{40}Ar)$, $\delta_\Omega(^{132}Xe/^{40}Ar)$, and $\delta_\Omega(^{22}Ne/^{40}Ar)$ simulated in the mid-latitudes at the ground surface and an altitude of 35 km. The values of $\delta_\Omega$ increased monotonically at the surface (decreased in the case of $\delta_\Omega(^{22}Ne/^{40}Ar)$) accompanied by seasonal cycles. In contrast to the ground surface, $\delta_\Omega$ values decreased (increased in the case of $\delta_\Omega(^{22}Ne/^{40}Ar)$) in the stratosphere. There was hence an increase of the difference between the stratosphere and ground surface. Table 3 summarizes the rate of change calculated by fitting a linear function to each $\delta_\Omega(X/Y)$. The long-term rates of change simulated at the ground surface were small for the $\delta_\Omega(X/Y)$ of the isotopes of Kr and Xe and for $\delta_\Omega(^{22}Ne/^{40}Ar)$ compared to the precisions of the observations. Such small changes would be difficult to detect at present. If the changes were monitored over a long period of time and the measurements were adequately precise, detectability of trends would be highest in the order $\delta_\Omega(^{40}Ar/^{28}N_2) > \delta_\Omega(^{40}Ar/^{36}Ar) > \delta_\Omega(^{34}O_2/^{32}O_2) > \delta_\Omega(^{84}Kr/^{40}Ar)$. In fact, Ishidoya et al. (2021) have estimated the effect of a change of stratospheric circulation on $\delta_\Omega(^{40}Ar/^{28}N_2)$ at the ground surface, and they have suggested that it has a significant influence on estimates of ocean heat content based on long-term $\delta(^{40}Ar/^{28}N_2)$ observations. If the precision of measurements is improved and/or high-frequency observations are made for other noble gases, it is likely that long-term trends will be detectable at the ground surface in future studies.

We have also conducted simulations of enhanced-RMC scenario so that the mean age of air decreased by 0.15 years decade$^{-1}$ at an altitude of 35 km over the northern mid-latitudes. The resulting trends in $\delta_\Omega$ at the ground surface and mid-stratosphere were the opposite of those of the weakened-RMC scenario, but their magnitudes were almost the same. Similar results have been also shown in trends simulated for $\delta(Ar/N_2)$ by Ishidoya et al. (2021).

Recent studies have shown that the shallow and deep branches of the BDC show different trends, and it is important to distinguish between them. Indeed, reanalysis data and 3-D model results have reported that the mean ages of air in the shallow and deep branches change differently (e.g. Garny et al., 2024). However, it is difficult to fully discuss the differences between the shallow and deep branches of the BDC using a 2-D model, this study simply examined the sensitivities of noble gas fractionations at the ground surface to the change in entire stratosphere as a first approach. We believe that the differences of changes between the shallow and deep branches of the BDC will be a future challenge, such as modeling noble gases using a 3-D model.

Recently, Ishidoya et al. (2025) have reported long-term trends of tropospheric $\delta(^{34}O_2/^{32}O_2)$ associated with the Dole–Morita effect. They continuously measured atmospheric $\delta(^{34}O_2/^{32}O_2)$ at Tsukuba, Japan, near the ground surface during 2013–2022 and found that the peak-to-peak amplitude of the average seasonal cycle was about 2 per meg and that there

was an increasing trend of +2.2 ±1.4 per meg decade$^{-1}$. The results of our simulations showed that the seasonal cycle
driven by fractionations in the upper atmosphere contributed 0.5 per meg to the peak-to-peak amplitude. However,
because the maximum of the simulated seasonal cycle was in late summer, the phase of the seasonal cycle was antiphase
to that observed by Ishidoya et al. (2025). The observed seasonal cycle of $\delta(^{34}O_2/^{32}O_2)$, which is driven by biological
processes, may therefore be slightly weakened by upper atmospheric fractionation. Furthermore, our results simulated
with the weakened-RMC scenario showed that the trend of $\delta_\Omega(^{34}O_2/^{32}O_2)$ at the ground surface could be +0.24 per meg
decade$^{-1}$. This rate of change is approximately 1/10 of the result observed by Ishidoya et al. (2025), but it may not be
negligible if detailed consideration is given to the long-term changes of the Dole–Morita effect.
**3.5 Sensitivity of kinetic fractionations to model dynamics**
Because spatiotemporal variations of the age of stratospheric air are governed only by stratospheric transport processes,
a study of those variations is an effective way to diagnose the dynamics of the stratosphere (e.g., Waugh and Hall, 2002;
Garny et al., 2024b). Many studies have therefore been conducted to compare the results of observations of the age of the
stratosphere with the results of numerical models. Climate models have predicted a long-term decrease of the age of air
in the northern mid-latitudinal mid-stratosphere. However, long-term balloon observations have not revealed such a
decreasing trend. The latest developments in studies of the age of air have been detailed by Garny et al. (2024b). At
present, observations of the age of air are inconclusive, and climate models have also not been able to adequately
reproduce the age of air. One major problem has been that many current climate models tend to underestimate the age of
air. In model calculations, the mean age of air can be decomposed into a residual circulation transit time (RCTT) and a
term characterized as "aging by mixing". The latter phenomenon is due to eddy mixing and recirculation, which Garny
et al. (2014) have argued is particularly important in the mid-latitude stratosphere. They have also suggested that the
strength of mixing is tightly coupled to the strength of the RMC. However, it is impossible to distinguish RCTT and
"aging by mixing" based on observations of age tracers alone. Gravitational separations and kinetic fractionations increase
with altitude and are similar to age of air in that respect, but they are thought to be physical quantities that are particularly
sensitive to vertical advection and mixing. This sensitivity reflects the fact that molecular diffusivities increase rapidly
with increasing altitude as atmospheric pressure decreases. It is therefore likely that the sensitivity of the fractionation of
noble gases differs from the sensitivity of the mean age of air to the changes of stratospheric transport processes and could
provide new constraints for numerical models in addition to the age of air.
In Section 3.4, we described the simulated results of noble gases in response to the scenario of a weakened RMC
(hereafter scenario A). In addition to this simulation, we performed a sensitivity test in which not only the RMC but also
the vertical and horizontal eddy diffusion coefficients, $K_{zz}$ and $K_{yy}$, were gradually weakened by 0.5 % year$^{-1}$ for the
entire atmosphere ("weakened-RMC&K" scenario, hereafter scenario B). We also performed calculations in which only
$K_{zz}$ was arbitrarily increased by a factor of 1.3 ("enhanced-$K_{zz}$ scenario", hereafter scenario C) because the Pe is directly
dependent on $K_{zz}$. It is therefore expected that the kinetic fractionations of noble gases will be significantly influenced by
a change of $K_{zz}$. Figure 13 shows the deviations of the mean age of air, $\delta_n(X/Y)$, as well as $\psi(X/Y)$ at an altitude of 35
km in the northern mid-latitudes simulated for each scenario. Here, deviations were equated to differences from the values
in the control run. The mean age of air at an altitude of 35 km increased because of weakening of the RMC and eddy
diffusion. The rates of increase were 0.15 and 0.20 years decade$^{-1}$ for scenario A and B, respectively. It has been reported
that an increase or decrease of the RCTT leads to an increase or decrease of "aging by mixing," and as a result, there is
almost a linear relationship between the RCTT and the age of air (Garny et al., 2014). The fact that the increase in the age
was greater in scenario B than in scenario A was generally consistent with the results of Garney et al. (2014).  In contrast,
the isotopic and elemental ratios were inversely correlated with the mean age of air. The fractionations increased with
increasing mean age in scenarios A and B, and vice versa in scenario C. Such anticorrelations between the mean age of
air and gravitational separations have already been reported in observations and model studies of the isotopic and
elemental ratios of the major components of the atmosphere (Ishidoya et al., 2013; Sugawara et al., 2018; Belikov et al.,
2019; Birner et al., 2020). In addition, we found that there was a tendency for the changes to be greater for light molecules
and smaller for heavy molecules. It is noteworthy that the value of $\psi(X/Y)$ responded differently in the A and B scenarios.
For example, $\psi(^{132}Xe/^{40}Ar)$ decreased in Scenario A and increased in Scenario B. Furthermore, the heavier the molecule,
the greater the change of $\psi(X/Y)$. This result implied that the kinetic fractionations of noble gases responded differently
to changes of the RMC and eddy diffusion. Furthermore, the results for scenario C, where $K_{zz}$ was increased, indicated
that the value of $\psi(X/Y)$ decreased significantly. The implication was that the kinetic fractionation was particularly
sensitive to vertical eddy diffusion. Although the deviations of $\psi(X/Y)$ shown here were so small that they could not be
detected with the current observational precision, the value of $\psi(X/Y)$ may be a useful tool in numerical models for
constraining stratospheric transport processes, especially in the case of vertical eddy diffusion.

648        In this regard, there is a discrepancy between the observed and modeled $\psi(X/Y)$ values. For example, the observed

$\psi(^{132}Xe/^{40}Ar)$ was $-0.61 \pm 0.03$ per meg (per meg)$^{-1}$, as shown in Table 1, whereas the simulated $\psi(^{132}Xe/^{40}Ar)$ was
approximately $-0.38$ per meg (per meg)$^{-1}$. Regardless of the scenarios, there was almost no change in the overestimation
of $\psi(X/Y)$ values. Model results showed that $\psi(^{132}Xe/^{40}Ar)$ dropped to around $-0.6$ per meg (per meg)$^{-1}$ at an altitude of
approximately 100 km where the molecular and eddy diffusions were competitive, but it was significantly overestimated
in the stratosphere. This overestimation of $\psi(X/Y)$ for noble gases in the model could be traced to the overestimation of
$r(X/Y)$ and was evident in the isotopic and elemental ratios of Ar, Kr, and Xe, as seen in Fig. 5b. The cause of this
overestimation is currently unclear. One of the possible causes would be an effect of thermal diffusion on the observed
data, as described before, an effect that was neglected in our model. However, we could not find clear evidence of
fractionations due to thermal diffusion in the stratosphere. Another possible cause could have been an unrealistic eddy
diffusion coefficient in the model. Although an enhancement of $K_{zz}$ could lead to depressions of $\psi(X/Y)$, as described
above, the age of air would simultaneously decrease. The result would be another contradiction. These results implied
that our sensitivity test using arbitrary changes of transport processes could not fully reproduce realistic $\psi(X/Y)$ values.
To resolve this problem, it will be necessary to carry out simulations with other models. In particular, simulations of the
gravitational separations and kinetic fractionations, in addition to the mean age of air, using modern climate models will
help to validate model dynamics. Model simulations that take thermal diffusion into account will also be needed in a
future study.

**3.6 Implications for ocean heat content and noble gas thermometry of mean ocean temperature**

Noble gases are extremely stable in the atmosphere, but they can be exchanged between the atmosphere and ocean because their solubilities in seawater change with seawater temperature variations. Because the temperature dependence of solubility is unique to each noble gas, the air-sea flux of noble gases due to changes in seawater temperature changes the elemental ratios of noble gases in the atmosphere (e.g. Keeling et al., 2004). Using this principle, the global mean ocean temperature (MOT) over the past several hundred thousand years has been reconstructed from $\delta(Kr/N_2)$, $\delta(Xe/N_2)$ and $\delta(Xe/Kr)$ of bubble air trapped in ice cores (e.g., Bereiter et al., 2018b; Shackleton et al., 2020; Haeberli et al., 2021). Ishidoya et al. (2021) reported the long-term variations of $\delta(Ar/N_2)$ observed at ground stations in 2012 – 2020 and discussed the contributions of secular changes in global ocean heat content (OHC) and BDC. They concluded that the effect of the BDC change on $\delta(Ar/N_2)$ at the ground surface cannot be ignored.

We extended the discussion of $\delta(Ar/N_2)$ by Ishidoya et al. (2021) to $\delta(Kr/Ar)$, $\delta(Xe/Ar)$, and $\delta(Ne/Ar)$, and roughly estimated the decadal-scale effects of increased OHC on the elemental ratios of noble gases, comparing them with the effects associated with BDC variations discussed in Section 3.4. The rate of change of $N_2$ and noble gases in response to an increase in OHC depends on seawater temperature, and the relative rates of change of $N_2$, Ar, Kr, Xe, and Ne per 100 ZJ (Zeta = $10^{21}$) of OHC are reported in Table 1 of Keeling et al. (2004). Assuming a seawater temperature of 10°C, the rates of change for $\delta(Ar/N_2)$, $\delta(Kr/Ar)$, $\delta(Xe/Ar)$, and $\delta(Ne/Ar)$ are 2.56, 6.12, 19.42, and –3.91 per meg (100 ZJ)$^{-1}$, respectively. The OHC value varies significantly depending on the depth of the ocean that is considered. Because we mainly focus on fluctuations over a relatively short timescale in this section, we used the annual OHC data reported by NOAA/NCEI up to a depth of 700 m (https://www.ncei.noaa.gov/access/global-ocean-heat-content/index.html, last access: September 18, 2025). Using this OHC data, the average rate of change in OHC over the 10-year period from 2010 to 2020 was calculated as 8.4 ZJ year$^{-1}$. Therefore, the temporal change rates of $\delta(Ar/N_2)$, $\delta(Kr/Ar)$, $\delta(Xe/Ar)$, and $\delta(Ne/Ar)$ associated with the increase in OHC during this period were estimated to be 2.2, 5.1, 16.3, and –3.3 per meg decade$^{-1}$, respectively. The rates of change in $\delta_\Omega$ caused by the change in BDC described in Section 3.4 (increase rate in Table 3) were comparable to the rates of change caused by increases in OHC. The relative magnitudes of the change rates due to the BDC change to those due to the OHC increase are 66, 71, 41, and 78 % for $\delta(Ar/N_2)$, $\delta(Kr/Ar)$, $\delta(Xe/Ar)$, and $\delta(Ne/Ar)$, respectively. This result suggests that OHC and BDC variations are essential for decadal-scale variations in the elemental ratios of noble gases.

It is interesting to determine how the BDC fluctuations can affect noble gases over the timescale of glacial-interglacial cycles. Fu et al. (2020) simulated BDC during the Last Glacial Maximum (LGM) using the Whole Atmosphere Community Climate Model (WACCM) and showed that the tropical upwelling during the LGM was weaker than that during the modern climate and that the mean age of air increased everywhere in the stratosphere during the LGM. We conducted an additional 2-D model simulation for a steady-state condition with a slow BDC in a simplified manner. The RMC in the model was weakened so that the mean age of air increased by 0.3 years (approximately 9 %) at an altitude of 35 km over the northern mid-latitudes when the model reached steady state after spin-up calculation during the 40-year period. As a result of this simulation, the changes in the annual average $\delta(Ar/N_2)$, $\delta(Kr/Ar)$, $\delta(Xe/Ar)$, and $\delta(Ne/Ar)$ at

the ground surface in southern high latitudes were 2.7, 7.0, 12.8, and –4.9 per meg, respectively, compared with those
before changing the RMC. If the BDC changes significantly with glacial-interglacial cycles, this may be recorded in the
noble gas elemental ratios in ice core samples, which may need to be considered when the past MOT is reconstructed
from noble gases.
**4 Conclusions**
Through continued high-quality sampling of stratospheric air and advances in gas analysis technology, this study revealed
for the first time the existence of small fractionations of noble gases in the stratosphere. The existence of gravitational
separation of major components of the stratosphere has been reported in our previous studies. It is nowadays recognized
that stratospheric gravitational separation is a tool that can be used to diagnose stratospheric transport processes (Garny
et al., 2024b). In addition to documenting gravitational separations, this study quantified the kinetic fractionations of
noble gases and revealed ways to apply them to the validation of transport processes in numerical models. This study
suggested that fractionations of noble gases due to molecular diffusion may respond in a unique way to long-term changes
of the RMC and eddy mixing and will differ from the response of the age of air. It is therefore likely that the molecular
diffusion of noble gases will be incorporated into numerical models such as chemical transport models and climate models
in future studies, and observational results will be used to provide constraints on the validity of the transport processes in
the models. Some models have already included molecular diffusion, and it would not be too difficult to incorporate it
into other models. An advantage in the case of simulations of gravitational separations and kinetic fractionations is that
the theory of molecular diffusion is well understood, and there is little ambiguity. Needless to say, the age of air is the
most powerful tool for diagnosing stratospheric transport processes. However, unlike the "clock-tracers" used in
numerical models, there are no truly ideal "clock-tracers" in observations. Even the mole fractions of $CO_2$ and $SF_6$, which
are often used as age tracers, are associated with several uncertainties, including nonlinear tropospheric variations and
mesospheric losses. Gravitational separations and kinetic fractionations have the great advantage that they are truly
governed by only transport processes.
At present, observations of noble gases in the atmosphere have been quite limited. In the future, observations of noble
gases will be needed not only in the stratosphere but also in the troposphere. The results of analyses of firn air and bubble
air in ice cores have been extremely useful for understanding the fractionations associated with molecular diffusion in the
atmosphere. Although there is a difference between firn air and the atmosphere where molecular diffusion and eddy
diffusion, respectively, are dominant, many concepts such as the age of air, age distributions (equivalent to "age spectra"
in atmospheric studies), gravitational separations, and kinetic fractionations (gravitational disequilibrium) are applicable
to both studies. Research on these will continue to progress in a complementary manner. From this perspective, it has
been speculated that thermal diffusion, another important equilibrium fractionation in firn air, probably also occurs in the
atmosphere. However, the depths at which gravitational separations, kinetic fractionations, and thermal diffusion in firn
air prevail differ to some extent. In contrast, all of these effects may be mixed throughout the atmosphere, and the
fractionation processes are more complex in the atmosphere than in firn air. This study could not clearly show the

influence of thermal diffusion, but because the sensitivities of thermal diffusion vary as a function of the gas species, this issue will likely be resolved by accumulating more observational data.

In this study, we improved the two-dimensional model to simultaneously calculate the mole fractions of many noble gases, and we were able to basically reproduce gravitational separations and kinetic fractionations in the stratosphere. However, simulations by numerical models are currently insufficient. A major problem is that we could not adequately reproduce the kinetic fractionations of noble gases. The fact that the two-dimensional model used in this study underestimated the strength of the kinetic fractionation suggested that observations of noble gases may provide constraints on the uncertainties of the transport processes in numerical models. At present, it is difficult to detect variations of noble gases elemental ratios in the troposphere caused by variations in stratospheric circulation, except for $\delta(^{40}Ar/^{28}N_2)$. However, when considering longer timescales such as glacial-interglacial cycles, variations in stratospheric circulation may have a significant impact on ice core data. It may also be necessary to incorporate thermal diffusion in future studies. Future research that uses more realistic three-dimensional models to attempt to reproduce not only the age of air, but also gravitational separations and kinetic fractionations of noble gases will be required.

**Data availability.** The observational data obtained by our balloon measurements are included as an electronic supplement to this manuscript.

**Supplement.** The supplement related to this article is available online at [the link will be implemented upon publication].

**Author contributions.** SS designed the study, conducted the balloon observations, and drafted the manuscript. IO and KK conducted the measurements of heavy noble gases at the National Institute of Polar Research. SI conducted the measurements of isotopic and elemental ratios of atmospheric major components and the balloon observations. SA, TN, SM, ST, and HH conducted the balloon observations. All authors approved the final manuscript.

**Competing interests.** The contact author has declared that none of the authors has any competing interests.

**Acknowledgements.** We deeply thank the Scientific Ballooning (DAIKIKYU) Research and Operation Group of the Institute of Space and Astronautical Science (ISAS), JAXA, Japan.

**Financial support.** This study was supported by Japan Society for the Promotion of Science KAKENHI grants (grant nos. 24K03070, 24H00762, 24H02345, 22H05006, 23K11396, and 23H00513).

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

933

**Tables and Figures**

Table 1. Summaries of the isotopic and elemental ratios measured in this study.

| Isotopic and elemental ratio | Reproducibility of measurements (per meg) | Difference of mass numbers ($\Delta m_{X,Y}$) (kg kmol$^{-1}$) | $\delta_n(X/Y)$[a] (per meg) | $r(X/Y)$ [b] (per meg (per meg)$^{-1}$) | $\psi(X/Y)$ [c] (per mge (per meg)$^{-1}$) | Std. dev. of r and ψ (per mge per meg$^{-1}$) | $m_{X,Y}$ [d] (kg kmol$^{-1}$) | $\mu_{X,Y}$ [e] |
|---|---|---|---|---|---|---|---|---|
| $\delta(^{29}N_2/^{28}N_2)$ | 2 | 1 | $-54 \pm 12$ | 1.00 | 0.00 | - | 28.49 | 1.0000 |
| $\delta(^{34}O_2/^{32}O_2)$ | 3 | 2 | $-52 \pm 9$ | 0.95 | -0.05 | 0.03 | 32.97 | 1.0115 |
| $\delta(^{40}Ar/^{28}N_2)$ | 8 | 12 | $-63 \pm 18$ | 1.20 | 0.20 | 0.05 | 32.94 | 0.9716 |
| $\delta(^{40}Ar/^{36}Ar)$ by NIPR | 4 | 4 | $-39 \pm 9$ | 0.72 | -0.28 | 0.04 | 37.89 | 0.9473 |
| $\delta(^{40}Ar/^{36}Ar)$ by AIST | 13 | | $-48 \pm 6$ | 0.86 | -0.14 | 0.05 | | |
| $\delta(^{40}Ar/^{38}Ar)$ | 7 | 2 | $-38 \pm 9$ | 0.70 | -0.30 | 0.03 | 38.97 | 0.9341 |
| $\delta(^{38}Ar/^{36}Ar)$ | 8 | 2 | $-40 \pm 11$ | 0.74 | -0.26 | 0.04 | 36.97 | 0.9591 |
| $\delta(^{86}Kr/^{82}Kr)$ | 15 | 4 | $-23 \pm 1$ | 0.42 | -0.58 | 0.04 | 83.95 | 0.4831 |
| $\delta(^{86}Kr/^{83}Kr)$ | 16 | 3 | $-23 \pm 4$ | 0.45 | -0.55 | 0.04 | 84.47 | 0.4816 |
| $\delta(^{86}Kr/^{84}Kr)$ | 10 | 2 | $-24 \pm 5$ | 0.51 | -0.49 | 0.06 | 84.99 | 0.4801 |
| $\delta(^{132}Xe/^{129}Xe)$ | 68 | 3 | $-16 \pm 4$ | 0.24 | -0.76 | 0.04 | 130.48 | 0.3196 |
| $\delta(^{136}Xe/^{129}Xe)$ | 150 | 7 | $-14 \pm 5$ | 0.29 | -0.71 | 0.04 | 132.41 | 0.3173 |
| $\delta(^{136}Xe/^{132}Xe)$ | 90 | 4 | $-18 \pm 5$ | 0.38 | -0.62 | 0.06 | 133.97 | 0.3155 |
| $\delta(^{84}Kr/^{40}Ar)$ | 57 | 44 | $-27 \pm 4$ | 0.50 | -0.50 | 0.05 | 54.19 | 0.6869 |
| $\delta(^{132}Xe/^{40}Ar)$ | 163 | 92 | $-21 \pm 5$ | 0.39 | -0.61 | 0.03 | 61.40 | 0.5809 |
| $\delta(^{22}Ne/^{40}Ar)$ | 163 | -18 | $-80 \pm 56$ | 1.19 | 0.19 | 0.28 | 28.39 | 1.4845 |

[a] The isotopic and elemental ratios normalized by the mass number difference and averaged at altitudes above 30 km. [b] The ratio of $\delta_n(X/Y)$ relative to $\delta(^{29}N_2/^{28}N_2)$ calculated from the observed data. [c] The excess value defined as $r(X/Y)$–1.

[d] Harmonic mean of the mass numbers for molecules X and Y. [e] The molecular diffusivity factor.

**Table 2.** Summaries of the molecular mass and diffusivities.

| molecule | $m_X$ [a] (kg kmol$^{-1}$) | $\sigma_X$ [b] | $m_{air,X}$ [c] (kg kmol$^{-1}$) | $D_{X,air}$ [d] (m$^2$ s$^{-1}$) | $D_{X,air}/D_{28N2,air}$ [e] |
|---|---|---|---|---|---|
| Air | 28.966 | 19.7 | - | - | - |
| $^{22}$Ne | 22 | 5.98 | 25.01 | 2.571E-05 | 1.495 |
| $^{28}$N$_2$ | 28 | 18.5 | 28.47 | 1.720E-05 | 1.000 |
| $^{29}$N$_2$ | 29 | 18.5 | 28.98 | 1.704E-05 | 0.991 |
| $^{32}$O$_2$ | 32 | 16.3 | 30.41 | 1.734E-05 | 1.009 |
| $^{34}$O$_2$ | 34 | 16.3 | 31.28 | 1.710E-05 | 0.994 |
| $^{36}$Ar | 36 | 16.2 | 32.10 | 1.691E-05 | 0.983 |
| $^{38}$Ar | 38 | 16.2 | 32.87 | 1.671E-05 | 0.972 |
| $^{40}$Ar | 40 | 16.2 | 33.60 | 1.653E-05 | 0.961 |
| $^{82}$Kr | 82 | 24.5 | 42.81 | 1.275E-05 | 0.742 |
| $^{83}$Kr | 83 | 24.5 | 42.94 | 1.273E-05 | 0.741 |
| $^{84}$Kr | 84 | 24.5 | 43.08 | 1.272E-05 | 0.739 |
| $^{86}$Kr | 86 | 24.5 | 43.34 | 1.268E-05 | 0.737 |
| $^{129}$Xe | 129 | 32.7 | 47.31 | 1.096E-05 | 0.637 |
| $^{132}$Xe | 132 | 32.7 | 47.51 | 1.093E-05 | 0.636 |
| $^{136}$Xe | 136 | 32.7 | 47.76 | 1.090E-05 | 0.634 |

[a] Mass number of molecule. [b] Diffusion volume parameter (Reid et al., 1987). [c] Harmonic mean of mass numbers
between the respective molecule and air. [d] Diffusivity of molecule X in air given by Eq. (12). These values are examples
at 1000 hPa and 273 K. [e] The ratio of molecular diffusivities relative to $D_{28N2,air}$. Note that these values are independent
of temperature and pressure.

**Table 3.** Values of $\delta_\Omega$ at 40°N on the ground surface simulated using the updated SOCRATES model.

| Isotopic and elemental ratio | Annual average of $\delta_\Omega$ [a] (per meg) | Seasonal amplitude [a] (per meg) | Increase rate [b] (per meg decade[-1]) |
|---|---|---|---|
| $\delta(^{29}N_2/^{28}N_2)$ | 2.4 | 0.3 | 0.12 |
| $\delta(^{34}O_2/^{32}O_2)$ | 4.6 | 0.5 | 0.24 |
| $\delta(^{40}Ar/^{28}N_2)$ | 27.7 | 3.1 | 1.41 |
| $\delta(^{40}Ar/^{36}Ar)$ | 8.8 | 1.0 | 0.45 |
| $\delta(^{40}Ar/^{38}Ar)$ | 4.4 | 0.5 | 0.22 |
| $\delta(^{38}Ar/^{36}Ar)$ | 4.5 | 0.5 | 0.23 |
| $\delta(^{86}Kr/^{82}Kr)$ | 6.5 | 0.7 | 0.33 |
| $\delta(^{86}Kr/^{83}Kr)$ | 4.9 | 0.5 | 0.25 |
| $\delta(^{86}Kr/^{84}Kr)$ | 3.3 | 0.3 | 0.16 |
| $\delta(^{132}Xe/^{129}Xe)$ | 4.3 | 0.4 | 0.21 |
| $\delta(^{136}Xe/^{129}Xe)$ | 10.0 | 1.0 | 0.50 |
| $\delta(^{136}Xe/^{132}Xe)$ | 5.7 | 0.6 | 0.28 |
| $\delta(^{84}Kr/^{40}Ar)$ | 72.2 | 7.8 | 3.66 |
| $\delta(^{132}Xe/^{40}Ar)$ | 132.3 | 14.0 | 6.66 |
| $\delta(^{22}Ne/^{40}Ar)$ | -50.0 | 5.6 | -2.56 |

[a] Annual average and seasonal peak-to-peak amplitude of $\delta_\Omega$ simulated for the control run. [b] Rate of secular increase of
$\delta_\Omega$ simulated for the weakened- residual mean circulation (RMC) scenario.


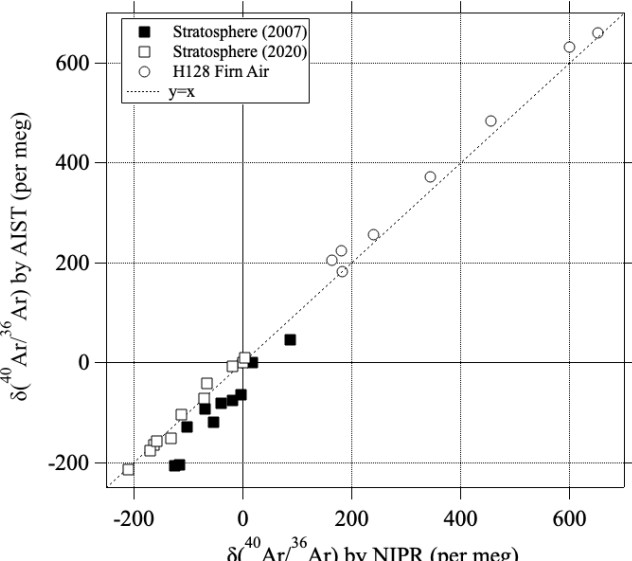


**Figure 1.** Comparisons of $\delta(^{40}Ar/^{36}Ar)$ values measured at the National Institute of Advanced Industrial Science and Technology (AIST) and National Institute of Polar Research (NIPR) for the stratospheric air samples (closed and open squares). The results measured for H128 firn air (Oyabu et al., 2025) are also shown by open circles. The linear function $y = x$ is shown by the dotted line.

959

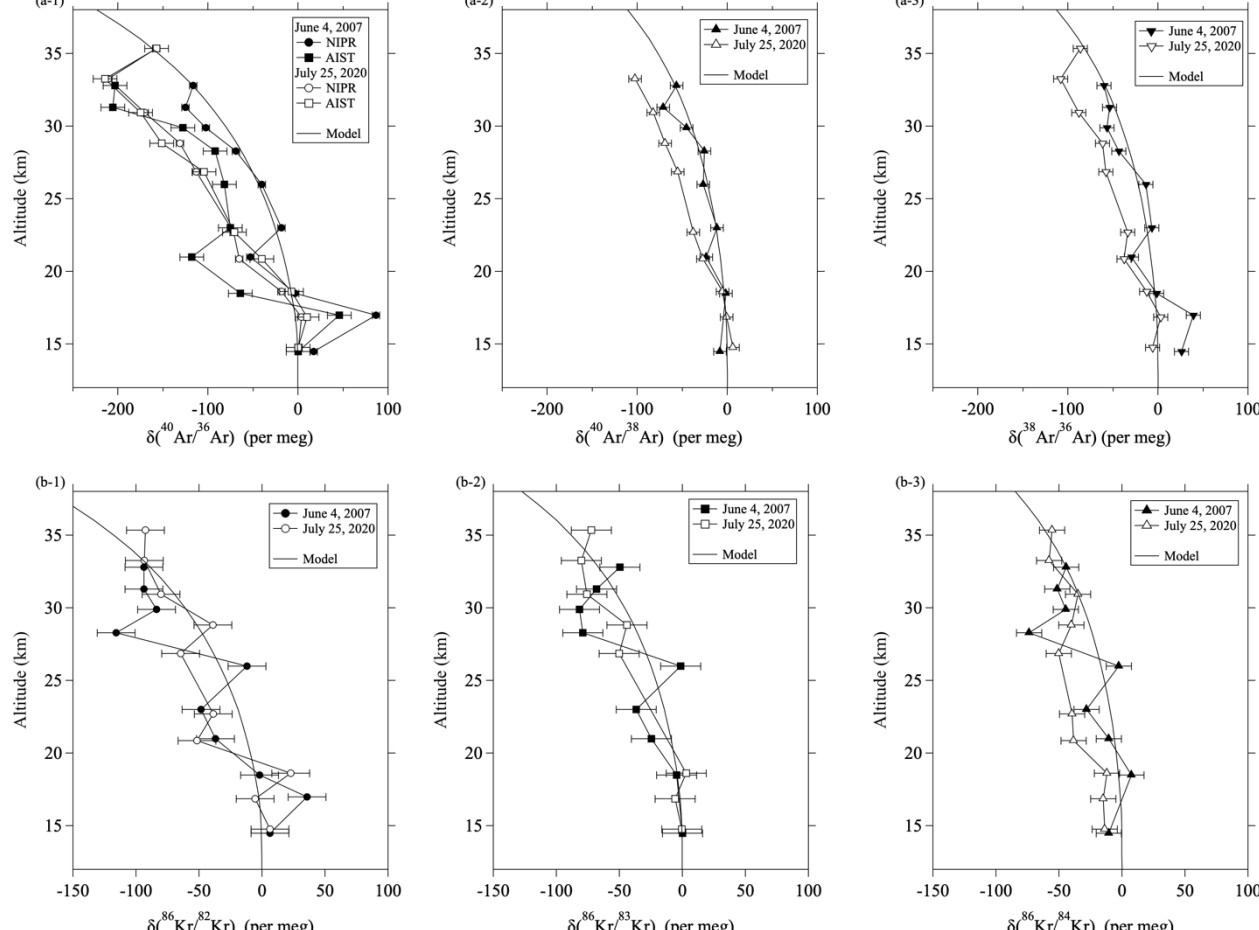

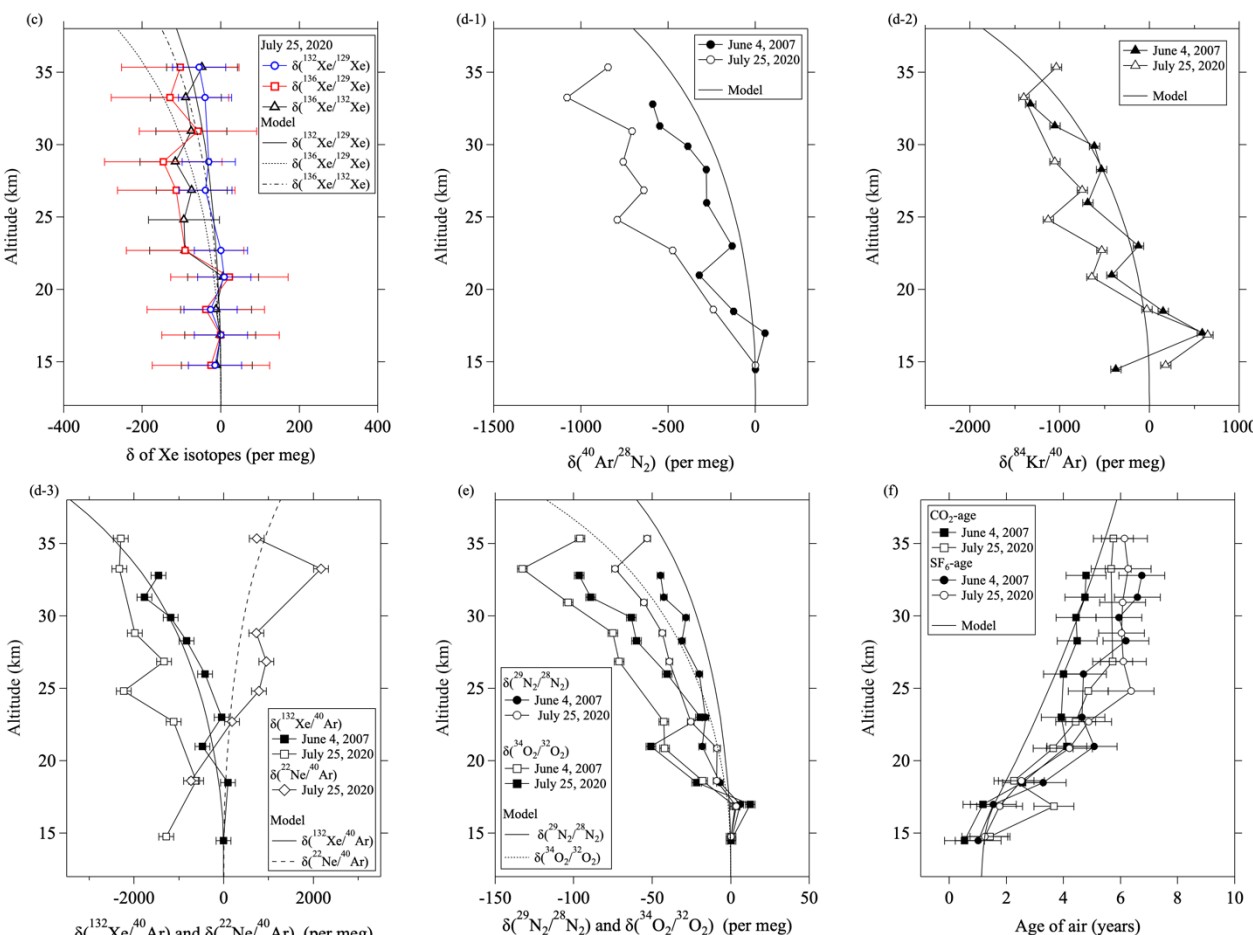

**Figure 2.** Vertical profiles of the isotopic ratios for (**a-1~3**) Ar, (**b-1~3**) Kr, and (**c**) Xe; (**d-1~3**) elemental ratios; (**e**) isotopic ratios of $N_2$ and $O_2$; and (**f**) age of air. Results of model simulations are shown by the various dotted and solid lines.

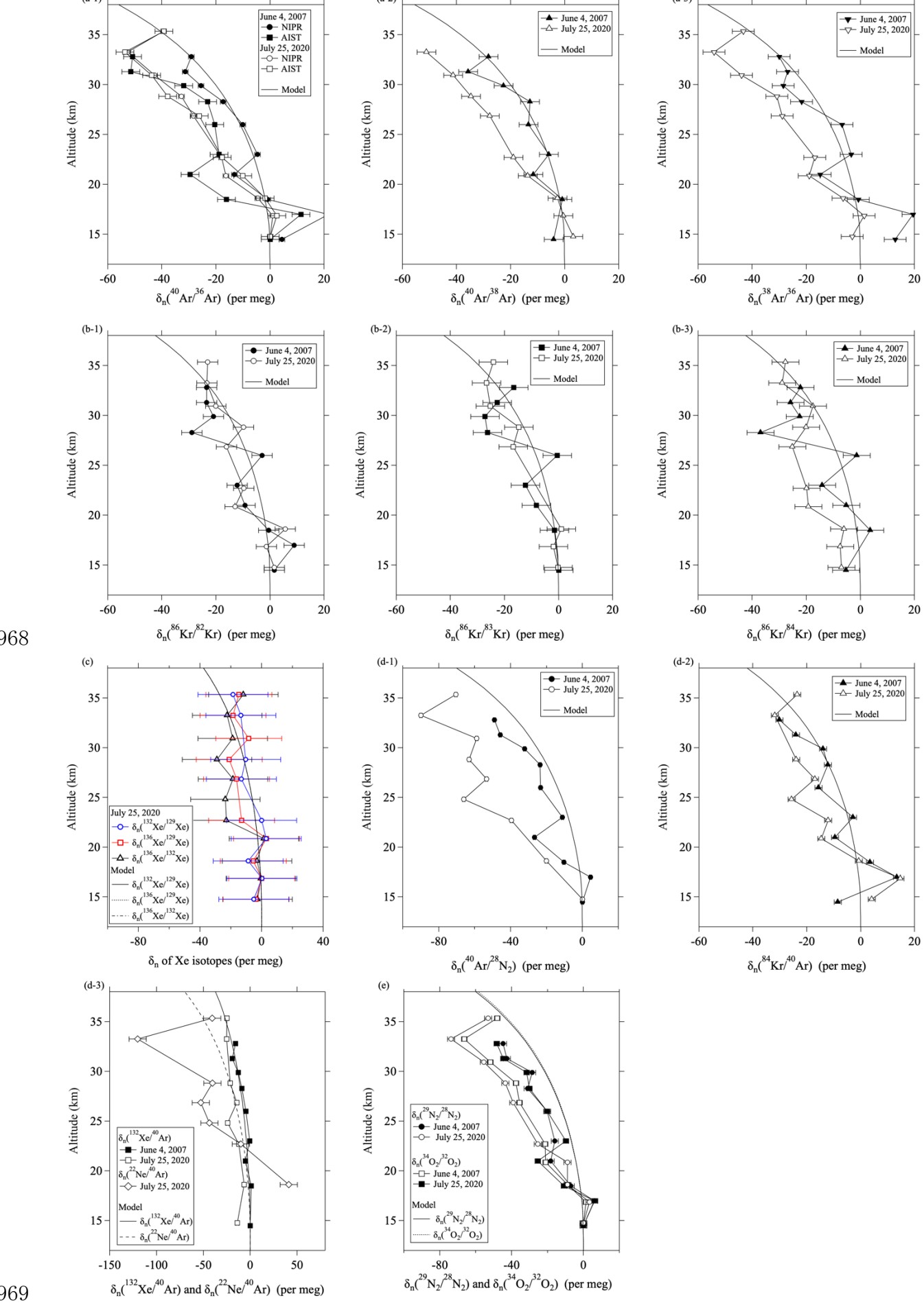



**Figure 3.** Same as Figure 2, but for the values normalized by the mass number differences ($\delta_n(X/Y)$).

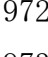

**Figure 4.** Plots of $\delta_n(X/Y)$ versus $\delta_n(^{29}N_2/^{28}N_2)$. Black lines are linear least-squares fits to the data. Results of model simulations are shown by black lines. The mass-dependent relationships (y = x) are shown by red lines.



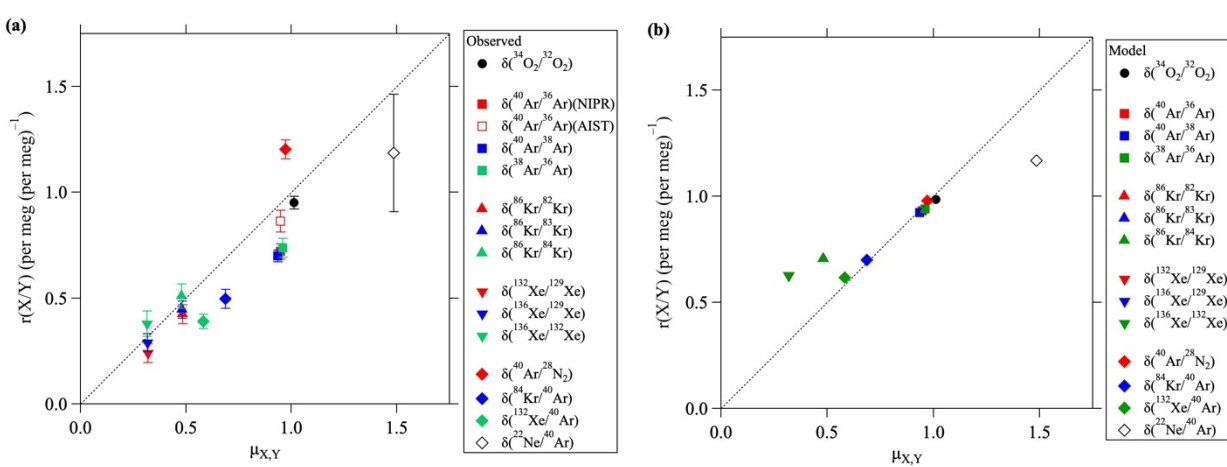


**Figure 5.** (**a**) Plots of the value of $r$(X/Y) versus the molecular diffusivity factor, $\mu_{X,Y}$. Dotted line shows the linear
function (y = x). (**b**) Same as (**a**), but for the results at mid-stratosphere over 40°N simulated by using a two-
dimensional model.


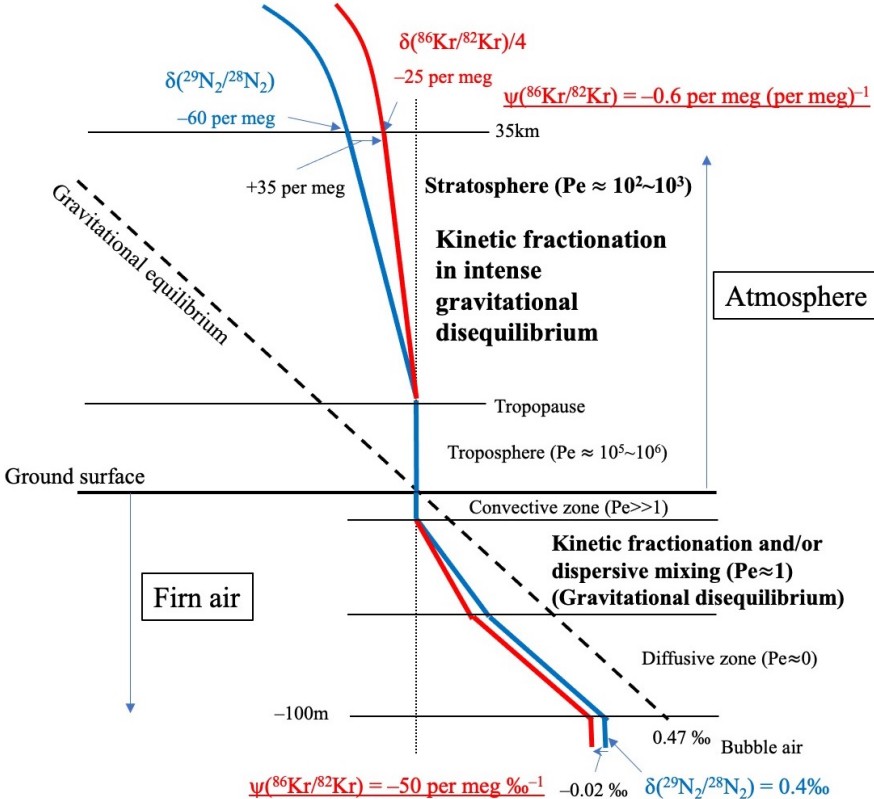


**Figure 6.** Schematic representation of vertical profiles of $\delta(X/Y)$ and the effects of kinetic fractionations in a firn and
the stratosphere. $\delta(^{29}N_2/^{28}N_2)$, $\delta(^{86}Kr/^{82}Kr)/4$, and its excess value, $\psi(^{86}Kr/^{82}Kr)$, are shown as an example of the heavy
nobles gases. Note that the scales of the vertical and horizontal axes differ between the firn and atmosphere.


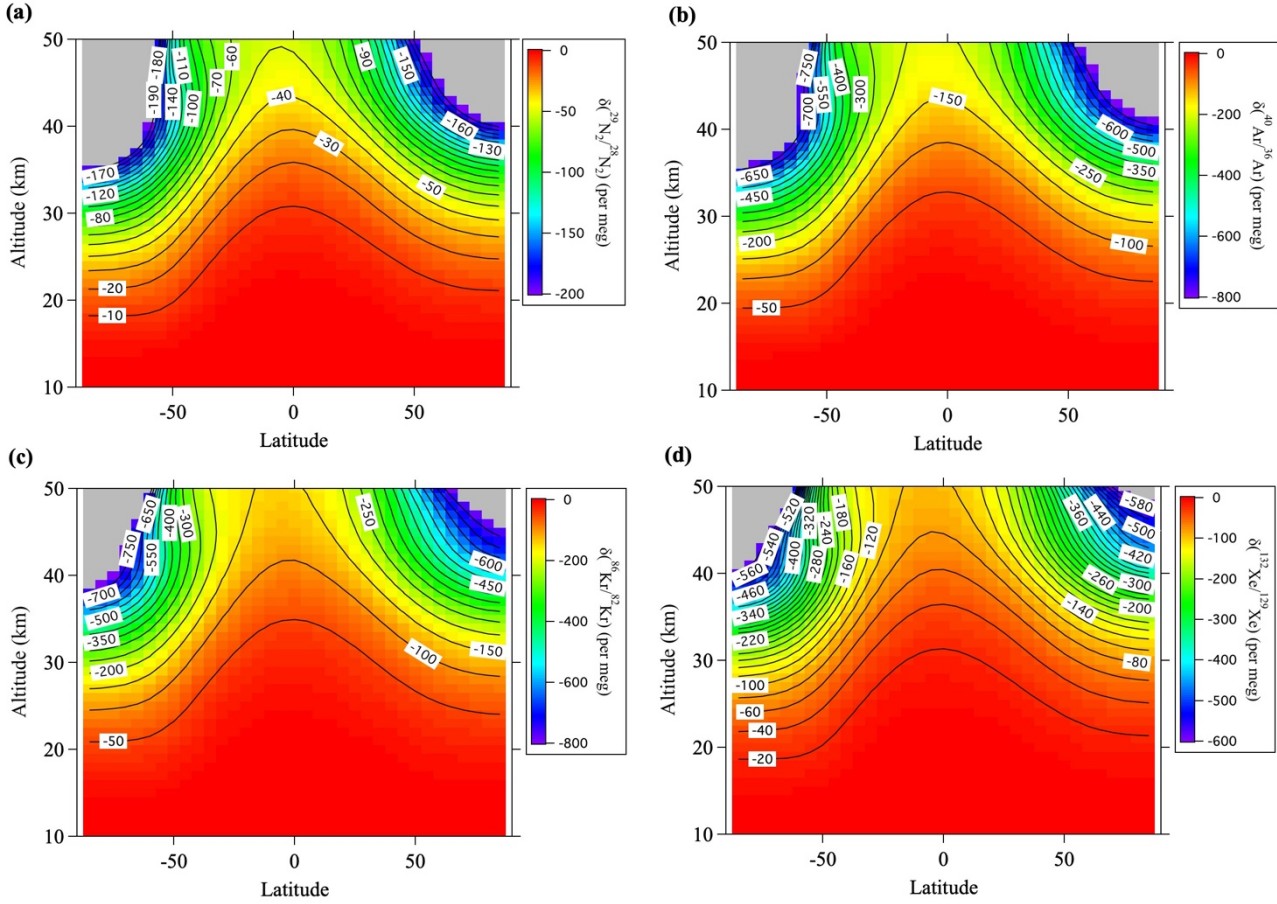


**Figure 7.** Average meridional distributions in June and July for (**a**) $\delta(^{29}N_2/^{28}N_2)$, (**b**) $\delta(^{40}Ar/^{36}Ar)$, (**c**) $\delta(^{86}Kr/^{82}Kr)$, and (**d**) $\delta(^{132}Xe/^{129}Xe)$, simulated using the updated SOCRATES model. Values lower than the lowest color contours are shown in gray.


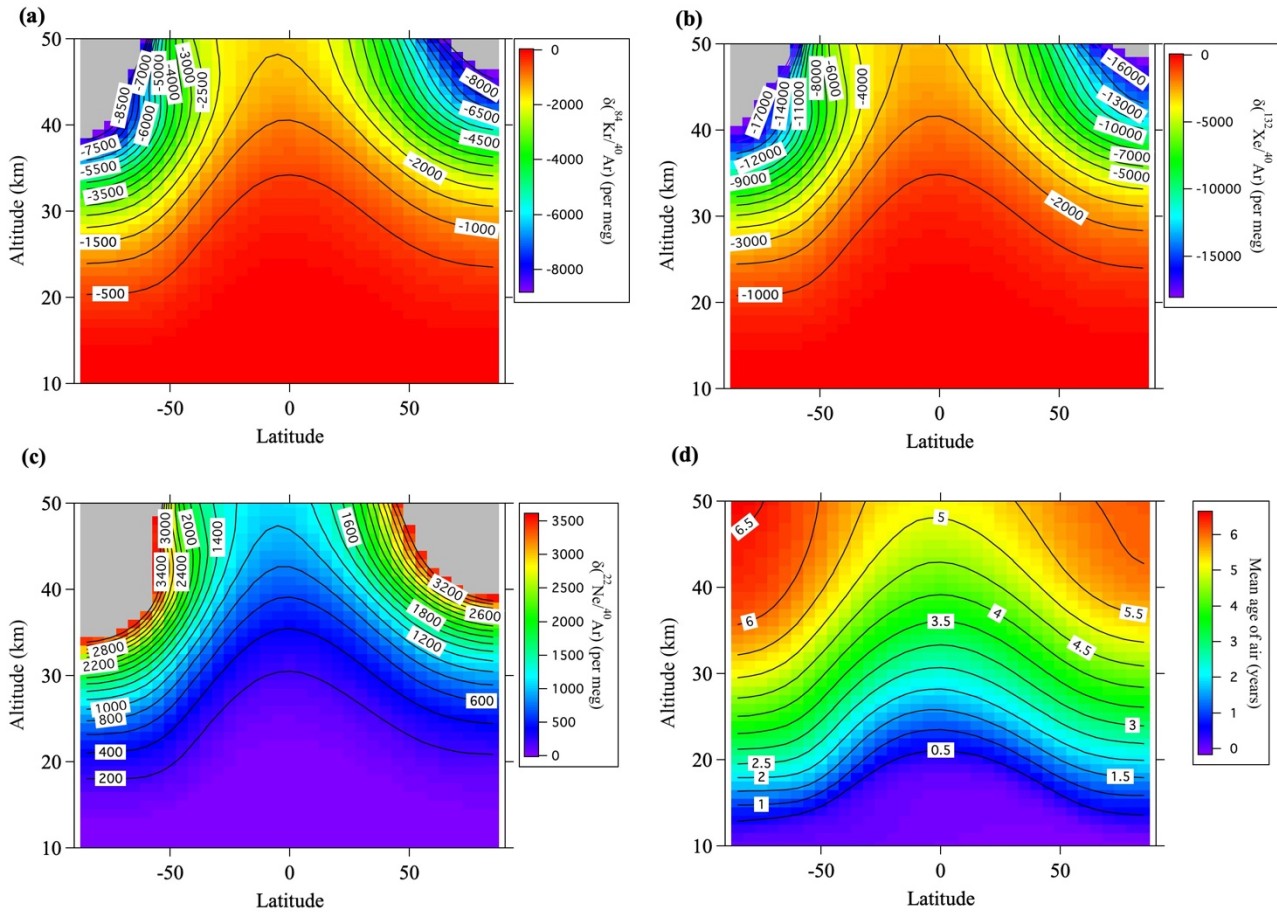


**Figure 8.** Same as Fig. 7, but for (**a**) $\delta(^{84}Kr/^{40}Ar)$, (**b**) $\delta(^{132}Xe/^{40}Ar)$, (**c**) $\delta(^{22}Ne/^{40}Ar)$, and (**d**) mean age of air.


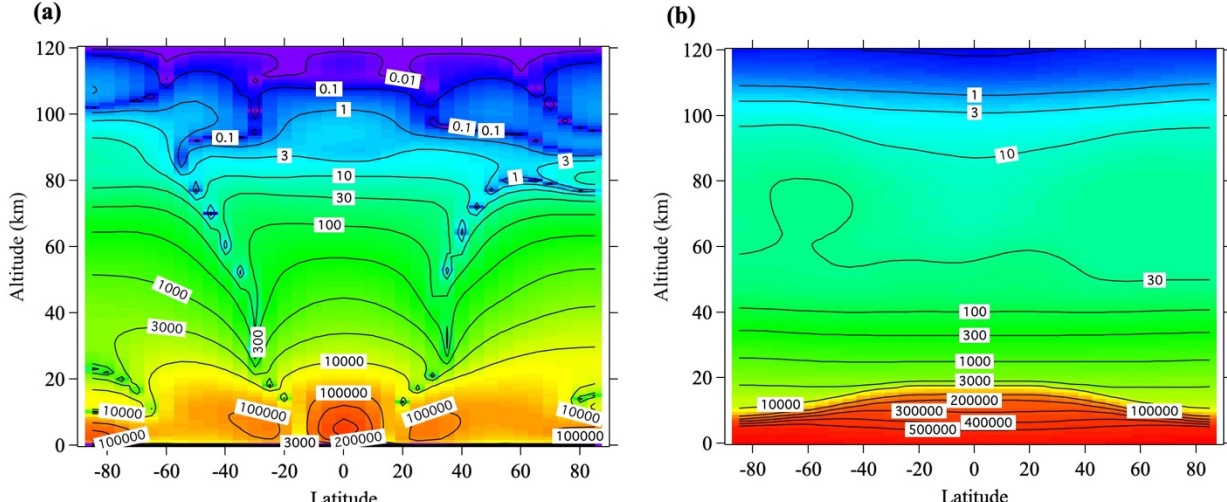

L003 **Figure 9.** Annual mean distributions of two components of the Péclet number (Pe), (**a**) $Pe_{28N2,w}$ and (**b**) $Pe_{28N2,K}$

L004 simulated for $^{28}N_2$ using the updated SOCRATES model.

L005

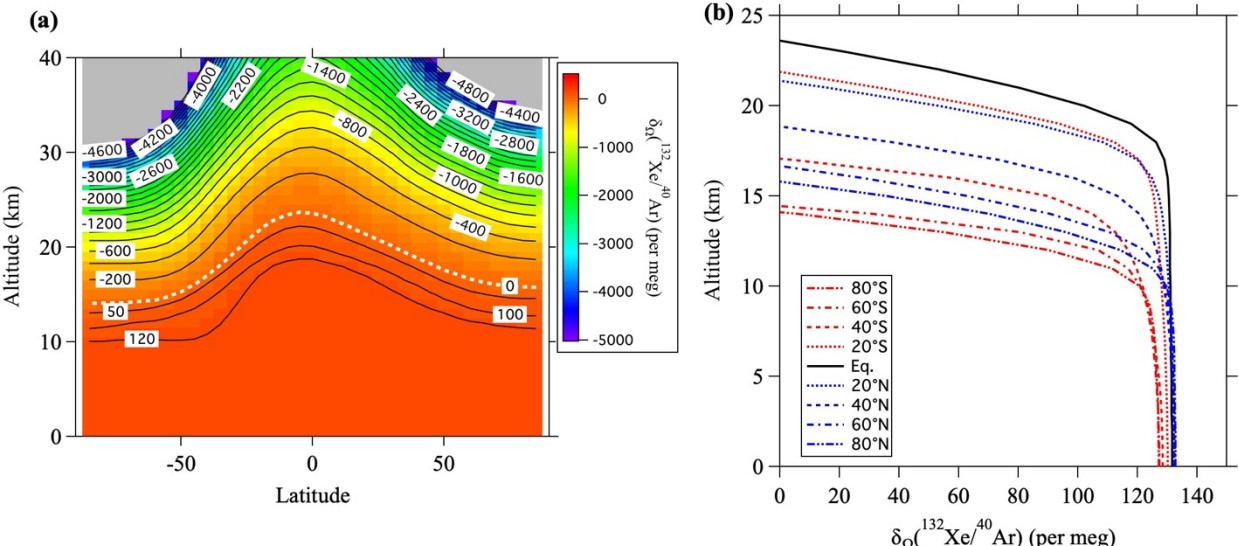

**Figure 10.** (**a**) Same as Fig. 8b, but for the annual average of $\delta_\Omega(^{132}Xe/^{40}Ar)$. The altitude at which $\delta_\Omega(^{132}Xe/^{40}Ar)$ is

zero is shown by a white dotted line. (**b**) Vertical distributions of the annual average of $\delta_\Omega(^{132}Xe/^{40}Ar)$ at latitudes from

80°S to 80°N. Only the regions where $\delta_\Omega(^{132}Xe/^{40}Ar)$ is positive are shown.

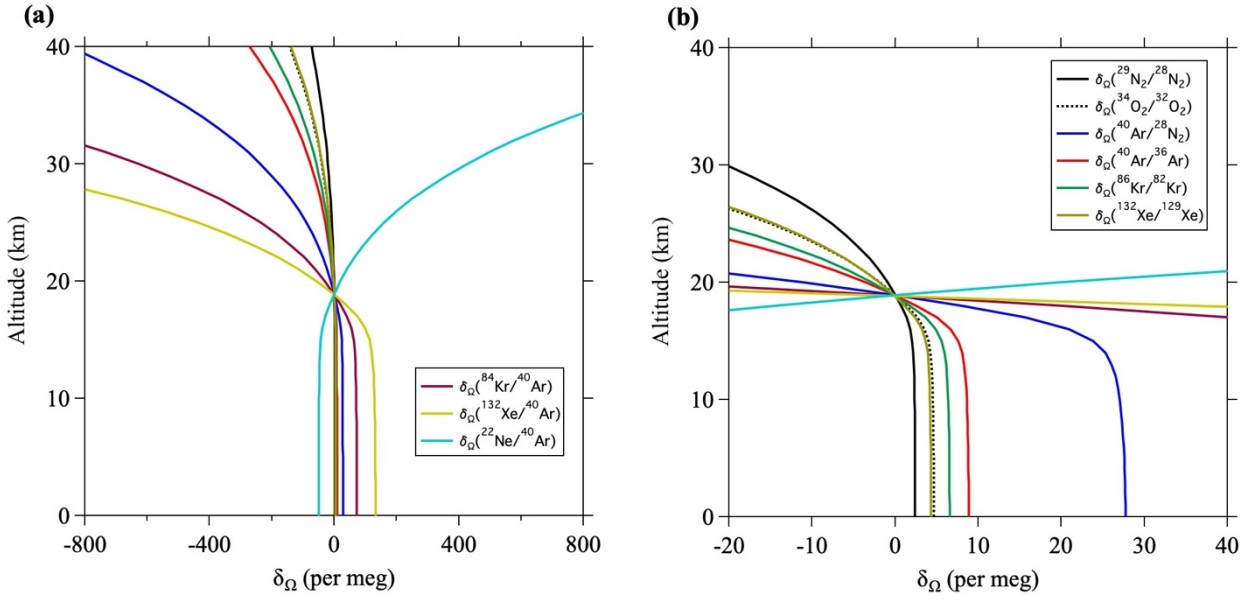

 **Figure 11.** (**a**) Vertical distributions of the annual average $\delta_\Omega$ at 40°N, calculated using the updated SOCRATES model.

 (**b**) Same as (**a**), but the horizontal axis is expanded close to zero.

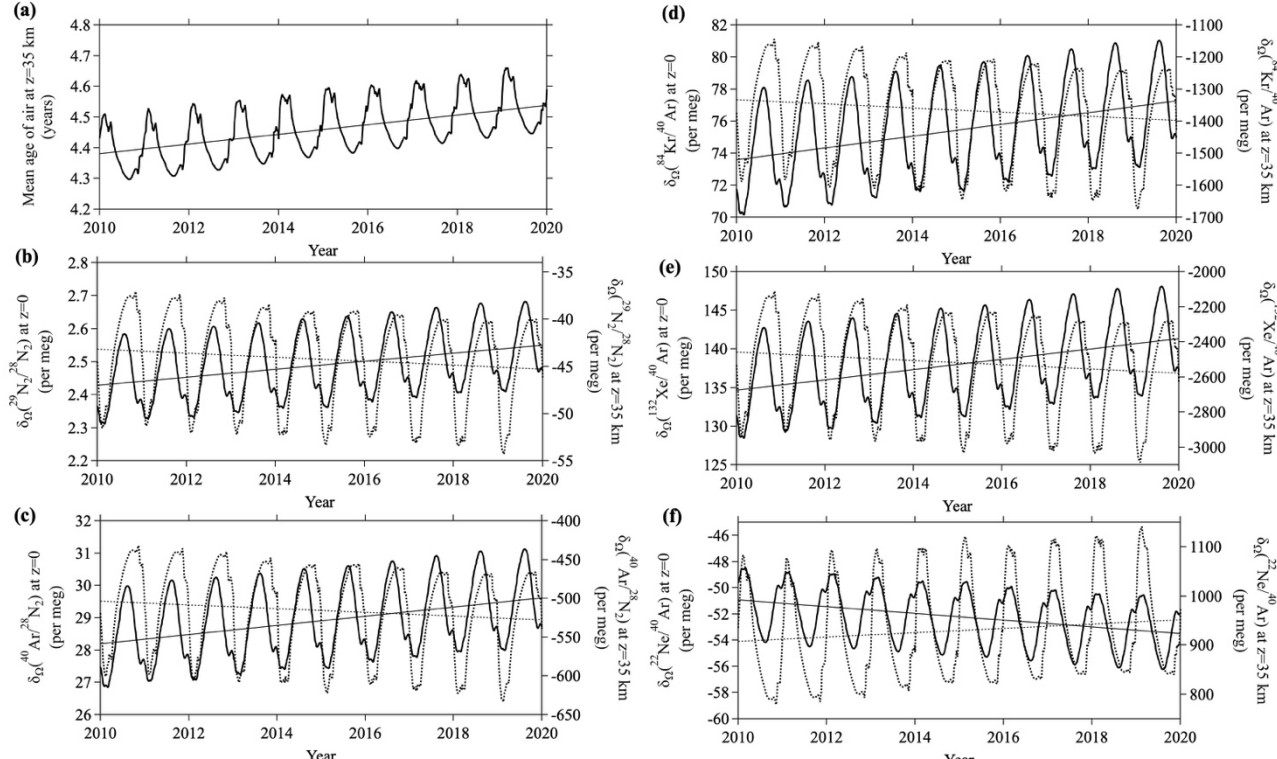

l019 **Figure 12.** Temporal variations of (**a**) mean age of air, and (**b**–**f**) δΩ(X/Y) values at 40°N simulated by using the updated

l020 SOCRATES two-dimensional model for the weakened-RMC scenario (see text). Thick solid lines and dotted lines in

l021 (**b**)–(**f**) show the values at the ground surface and at an altitude of 35 km, respectively. Linear lines denote secular

l022 trends obtained by applying linear regression analyses.

l023

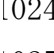

L026

**Figure 13.** Deviations of (**a**) mean age of air, (**b**) $\delta_n(X/Y)$, and (**c**) $\psi(X/Y)$ from the values simulated in the control run.

A, B, and C denote the weakened-RMC, weakened-RMC&K, and enhanced-$K_{zz}$ scenarios, respectively (see text).

Deviations are calculated from the annual mean at an altitude of 35 km over 40°N.