# Peer review of "Kinetic fractionation of noble gases in the stratosphere over Japan"

_EGUsphere, 2025_

## Referee Comment (RC1)

Kinetic fractionation of noble gases in the stratosphere over Japan, Sugawara et al., EGUsphere 2025

Main objective of manuscript: Evaluate the vertical distribution of noble gases (Kr, Xe, and Ne) over the stratosphere, and processes governing it.

Main approach: Measurements of noble gases elemental and isotopic composition at different stratospheric altitude collected using balloon borne cryogenic air samples over Japan. Combined with two dimensional atmospheric model simulation (SOCRATES).

Main results: Elemental ratios increase, and isotopic ratios decrease for all noble gases with increasing altitude.

Implication: Kinetic fractionation occurred in the stratosphere because of difference in molecular diffusivity, can be a new tool to diagnose stratospheric mass transport processes.

**Review**

Major Comments:

- Section 2.3: Mean age of air: this section was a little hard to follow and could be improved with more information related to the statements made in this paragraph. Below are examples of lines where more clarification would be very helpful:
    - Line 178-179: "*certain relationship between the gravitational separation of the major atmospheric components and the mean age of stratospheric air*".
        - What the relationship is, is unknown to the reader, and should be stated clearly.

    - Line 180-183: "*We also measured the mole fractions of CO2 and SF6 in our stratospheric air samples…. Because this method of estimation has already been described in previous studies… only a brief description is presented here.*"
        - It should be stated clearly why CO2 and SF6 are species specifically chosen to measure the age of air. That information is currently not available here and may not be known to all.

    - Lines 188-200: "*The mean age was estimated…. ratio of moments to be 1.25years*"
        - This section is mostly unclear to me, although my expertise in this is limited. Some clarifications would be helpful though. First, along with clarifying why CO2 and SF6 mole fractions are chosen specifically, clarifying what the convolution method is, would also be very helpful. Specifically, clarifying what the tropospheric reference curve and the hypothetical age spectrum represent physically/temporally.
        - Additionally, describing where the ratio of moments relationship comes from and its physical significance would be helpful for the reader. It seems important in this context. Also, why the 1.25years is the choice made here should be explained. The authors mention

that this was reported in a previous study but, clarifying why they think that this is accurate to use, would be helpful.

- Section 3.1:
  - General suggestion on Figure 2: This figure is an important figure for this manuscript as it shows the measurement results of this study. The figure is hard to read and decipher both in print and on the computer. The legends are too small to read, and it would be more useful to use different color/bigger markers to show these findings. The mode result lines are also hard to distinguish in the midst of all the measurement results.

    I would recommend improving the readability of this figure, as it reports the major findings of this study. Additionally, it would be useful to have a visual representation of the uncertainty in these measurements, either as errorbars in the figures or just as a legend that represents the uncertainty of the data.

  - Line 213-215: "*relative to the values observed in the lowest layers of the balloon observations…. negligibly small*"

    Clarify why this would be the case.
  - Line 216: "*clearly apparent... decreased with increasing altitude*".

    Although this statement is broadly true and the data shown in figures 2 &3 clearly highlight that, the elemental and isotopic composition of all 3 noble gases in figures 2&3 highlight excursions around specific altitudes. This is also apparent in all the data. For Xe, its between 20-25 km, Kr 25-30km, and Ar 15-20km.

    This does not seem included in the first section of the results, although they seem quite consistent. It would be useful to also note these in the results, and including them in the ensuing discussions.
  - Line 228: "*The fluctuations of the Xe isotopic ratios were irregular and larger than those of the Ar and Kr isotopic ratios*"

    Why?

- Section 3.2: I thought this section is really well written and clear, and easy to understand and follow. Lines 379-385 particularly do a good job of summarizing simply the key findings of this section.

- Section 3.4:
  - Lines 505-507: " *if Brewer Dobson circulation strengthened over time… reduce the mean age of stratospheric air.*"

    Does this change refer to the BDC as a whole? If my understanding is correct, there is shallow arm of the BDC that causes mass exchange between the statosphere and troposphere across the 380 K isentrope, and a deeper arm of the BDC that exchanges mass with greater altitudes of the atmosphere.

    Does the model result distinguish between these? Would a difference between this matter? Presumably if only the shallow branch changed, it would have a different magnitude of impact on the troposphere than if the deeper branch changed? Would it be possible to disentangle this information from the model results? There may be other existing 3-D modeled results (for e.g. https://link.springer.com/article/10.1007/s00382-

006-0162-4 ) that could shed more insights on this. This would be relevant and helpful information to have I believe.

o   Lines 511-520: It is unclear to me why the authors chose to only simulate increased mean age of air in this study, if it is unclear whether the mean mid-stratospheric age in increasing or decreasing. It would make more sense to simulate both increased and decreased mean ages, or explain more clearly why this choice was made.

o   Line 528-530: "*significant influence on estimates of ocean heat content..*"

Is there any way to get some first order constraints on this? For e.g., would it be possible to make a statement like "if stratospheric circulation changed by X%, we would expect it to have Y% impact on estimates on long term ocean heat content" or something along these lines? This seems like a very important constraint to have on this proxy. Noble gas thermometry for mean ocean temperatures from ice cores (for e.g.: [https://www.researchsquare.com/article/rs-5610580/v1](https://www.researchsquare.com/article/rs-5610580/v1)) seem to fundamentally assume no other processes are affecting it. If there is indeed an effect from circulation, it would be very important to know how large this effect could be, and how that translates to effects on MOT reconstructions.

---

## Author Comment (AC1)

**Author Responses to the Referee #1 Comments**

Thank you very much for your significant and useful comments on the paper "Kinetic fractionation of noble gases in the stratosphere over Japan" by Sugawara et al. We have revised the manuscript, considering your comments and suggestions. Details of our revision are as follows. The line numbers denote those of the revised manuscript.

*Section 2.3:*
*Mean age of air: this section was a little hard to follow and could be improved with more information related to the statements made in this paragraph. Below are examples of lines where more clarification would be very helpful:*

*Line 178-179: "certain relationship between the gravitational separation of the major atmospheric components and the mean age of stratospheric air".*
   *What the relationship is, is unknown to the reader, and should be stated clearly.*

We have deleted the sentences, "Previous studies have shown that there is a certain relationship between the gravitational separation of the major atmospheric components and the mean age of stratospheric air (Ishidoya et al., 2013; Sugawara et al., 2018; Belikov et al., 2019; Birner et al., 2020).", and replaced as follows:

Lines 182-189:

"Previous studies have shown that the gravitational separation of the major atmospheric components strengthens with increasing altitude ($\delta$ values decrease with increasing altitude), and the mean age of stratospheric air increases simultaneously. Therefore, it is known that the vertical distributions of $\delta(^{29}N_2/^{28}N_2)$ and mean age of air show anti-correlations (Ishidoya et al., 2013; Sugawara et al., 2018). These correlations have also been reproduced by 3-dimensional model studies (Belikov et al., 2019; Birner et al., 2020). Furthermore, an anti-correlation in the interannual variations of the gravitational separation and the mean age of air has been observed in the northern mid-latitude mid-stratosphere (Ishidoya et al., 2013). This means that gravitational separation becomes stronger when the relevant stratospheric air becomes older."

*Line 180-183: "We also measured the mole fractions of CO2 and SF6 in our stratospheric air samples…. Because this method of estimation has already been described in previous studies… only a brief description is presented here."*
   *It should be stated clearly why CO2 and SF6 are species specifically chosen to measure the*

*age of air. That information is currently not available here and may not be known to all.*

We have deleted the sentences, "We also measured the mole fractions of $CO_2$ and $SF_6$ in our stratospheric air samples. These mole fractions are often used to estimate the mean age of stratospheric air.", and added new sentences as follows:

Line 189-193:

"The mean age of air has been estimated based on observation data of inert trace gases in the stratosphere. If an inert tracer shows a linear trend in troposphere, the time lag between tropospheric and stratospheric mole fractions is the mean age of air. The mole fractions of $CO_2$ and $SF_6$ have been widely used for this purpose, because both species are almost inert in the stratosphere and show monotonous increase trends in troposphere. Therefore, we measured the mole fractions of $CO_2$ and $SF_6$ in our stratospheric air samples and calculated the mean age of air as described below."

*Lines 188-200: "The mean age was estimated…. ratio of moments to be 1.25years"*

*This section is mostly unclear to me, although my expertise in this is limited. Some clarifications would be helpful though. First, along with clarifying why CO2 and SF6 mole fractions are chosen specifically, clarifying what the convolution method is, would also be very helpful. Specifically, clarifying what the tropospheric reference curve and the hypothetical age spectrum represent physically/temporally.*

*Additionally, describing where the ratio of moments relationship comes from and its physical significance would be helpful for the reader. It seems important in this context. Also, why the 1.25years is the choice made here should be explained. The authors mention that this was reported in a previous study but, clarifying why they think that this is accurate to use, would be helpful.*

We have totally revised mean age section 2.3 and added more detailed sentences about age spectrum, convolution method, and ratio of moments as follows:

Lines 201-203:

"The convolution method is a method for determining the mean age by calculating the convolution of the age spectrum and a tropospheric reference curve and comparing it with the observed value."

Lines 208-209:

"The age spectrum is defined as the statistical probability of individual transit times of different air parcels arrived at a certain place in stratosphere after air parcels intruded into the stratosphere through the tropical upper troposphere. The age spectrum naturally changes over time, but it was ignored here."

Lines 212-219:

"This function is known to be the Green's function of one-dimensional advective diffusion differential

equation (Hall and Plumb, 1994). This function will be calculated by assuming the spectral width ($\Delta$) and mean age ($\Gamma$). The mean age is determined by successively calculating the convolutions of equation (3) while varying the value of $\Gamma$ and comparing $x(\Gamma, t)$ with the observed mole fractions. However, $\Delta$ is still unknown. $\Delta$ represents the effect of mixing process in atmospheric transport, and it is expected that the larger the mean age, the greater the effect of mixing and the $\Delta$ will be. Therefore, $\Delta$ is assumed to be given by the relationship $\Delta^2/\Gamma$ = constant (years). This value, called the ratio of moments, is suggested by Hall and Plumb (1994) from the results of a stratospheric AGCM. A ratio of moments of 0.7 years has been widely used in previous studies (e.g. Engel et al. 2002)."

Lines 221-223:

"Fritsch et al. (2020) have reported that the mean age calculated from a virtual tracer with a linear trend using a numerical model is in good agreement with observed values when the ratio of moments value of 1.25 is assumed."

*Section 3.1:*

*General suggestion on Figure 2: This figure is an important figure for this manuscript as it shows the measurement results of this study. The figure is hard to read and decipher both in print and on the computer. The legends are too small to read, and it would be more useful to use different color/bigger markers to show these findings. The mode result lines are also hard to distinguish in the midst of all the measurement results.*

*I would recommend improving the readability of this figure, as it reports the major findings of this study. Additionally, it would be useful to have a visual representation of the uncertainty in these measurements, either as errorbars in the figures or just as a legend that represents the uncertainty of the data.*

We have replaced Figures 2, 3, 4, and 13 to improve the readability. Error bars have been added to all the vertical profiles in Figures 2 and 3.

*Line 213-215: "relative to the values observed in the lowest layers of the balloon observations…. negligibly small"*

       *Clarify why this would be the case.*

Our aircraft observations have reported that there is no significant vertical gradient of the isotopic and elemental ratios of atmospheric major compositions from the surface to the tropopause (Ishidoya et al., 2008a). We have added sentences about that point and relevant references.

Lines 242-250:

"Although there are very few observations of the isotopic and elemental ratios of atmospheric major components in the upper troposphere, aircraft observations have reported that there is no significant vertical gradient of $\delta(^{29}N_2/^{28}N_2)$ and $\delta(^{34}O_2/^{32}O_2)$ from the surface to near the tropopause (Ishidoya et al., 2008a). On the other hand, Bent (2014) observed $\delta(^{40}Ar/^{28}N_2)$ in air samples obtained by the HIAPER Pole-to-Pole Observations (HIPPO) project and reported a large vertical gradient in troposphere. However, such vertical gradient could not be explained by a 1-D atmospheric diffusion model, and it was unclear whether it is either natural or artificial. As will be discussed later, our results of 2-D model also show very small differences between the surface and the upper troposphere. Because the air samples at the lowest layer were collected below the tropopause in our balloon observations, the differences of the isotopic and elemental ratios between the ground surface and tropopause should be negligibly small."

*Line 216: "clearly apparent... decreased with increasing altitude".*

*Although this statement is broadly true and the data shown in figures 2 &3 clearly highlight that, the elemental and isotopic composition of all 3 noble gases in figures 2&3 highlight excursions around specific altitudes. This is also apparent in all the data. For Xe, its between 20-25 km, Kr 25-30km, and Ar 15-20km. This does not seem included in the first section of the results, although they seem quite consistent. It would be useful to also note these in the results, and including them in the ensuing discussions.*

As you pointed, we have added a paragraph of irregular variations in vertical distributions of isotopic and elemental ratios as follows:

Lines 271-279:

"As seen in Figure 2 and 3, there are irregular fluctuations of the isotopic and elemental ratios in the vertical distributions and some of them occur synchronously within the same gas species. Ar isotopic ratios, $\delta(^{40}Ar/^{36}Ar)$, $\delta(^{40}Ar/^{38}Ar)$ and $\delta(^{38}Ar/^{36}Ar)$ observed in 2007 showed irregularly low values at altitude of 21 km. Kr isotopic ratios observed also showed similar variations at altitude of 26 km in 2007. Xe isotopic ratios below 21 km also showed similar variations, although their statistical significance is low. Unfortunately, the cause of these irregular variations is not clear at present. Because the irregular variations are not common to all gas species, it is likely that there were factors that had different effects on the gas species. The causes are not necessarily natural, and the possibility of small fractionations during sample air pretreatments cannot be ruled out. This issue, along with irregular variations in the isotopic and elemental ratios of the atmospheric major components, remains to be solved in the future."

*Line 228: "The fluctuations of the Xe isotopic ratios were irregular and larger than those of the Ar*

*and Kr isotopic ratios"*

*Why?*

We have replaced Fig. 2 and 3 as described above. This shows that the error bars in the Xe isotopic ratios are larger than the vertical variations. Taking these into consideration, the descriptions of Xe isotope ratios have been changed as follows:

Lines 261-265

We have deleted following sentences:

"The isotopic ratios of Xe—$\delta(^{132}Xe/^{129}Xe)$, $\delta(^{136}Xe/^{129}Xe)$, and $\delta(^{136}Xe/^{132}Xe)$—depended on mass number differences in a similar way. The fluctuations of the Xe isotopic ratios were irregular and larger than those of the Ar and Kr isotopic ratios.",

and replaced with,

"Because the uncertainties in the analysis of Xe isotopic ratios —$\delta(^{132}Xe/^{129}Xe)$, $\delta(^{136}Xe/^{129}Xe)$, and $\delta(^{136}Xe/^{132}Xe)$— were much larger compared to their vertical changes (Fig. 2c), their vertical gradients were not very significant. However, the larger the mass number difference, the larger the decreasing with altitude, and a mass-dependent relationship similar to that observed for Ar and Kr isotopic ratios was barely observed.".

*Section 3.2:*

*I thought this section is really well written and clear, and easy to understand and follow. Lines 379-385 particularly do a good job of summarizing simply the key findings of this section.*

It's an honor to hear that from you.

*Section 3.4:*

*Lines 505-507: " if Brewer Dobson circulation strengthened over time… reduce the mean age of stratospheric air."*

*Does this change refer to the BDC as a whole? If my understanding is correct, there is shallow arm of the BDC that causes mass exchange between the stratosphere and troposphere across the 380 K isentrope, and a deeper arm of the BDC that exchanges mass with greater altitudes of the atmosphere.*

*Does the model result distinguish between these? Would a difference between this matter? Presumably if only the shallow branch changed, it would have a different magnitude of impact on the troposphere than if the deeper branch changed? Would it be possible to disentangle this information from the model results? There may be other existing 3-D modeled results (for e.g.*

*https://link.springer.com/article/10.1007/s00382- 006-0162-4 ) that could shed more insights on this. This would be relevant and helpful information to have I believe.*

We completely agree with your comment. Ideally, we should have calculated the changes in the shallow and deep branches of BDC separately, but we simplified the scenario as the first approach to noble gases. We believe that the difference between the shallow and deep branches is a topic for future study. We have added a description of this.

Lines 585-591:

"Recent studies have shown that the shallow and deep branches of the BDC show different trends, and it is important to distinguish between them. Indeed, reanalysis data and 3-D model results have reported that the mean ages of air in the shallow and deep branches change differently (e.g. Garny et al., 2024). However, it is difficult to fully discuss the differences between the shallow and deep branches of the BDC using a 2-D model, this study simply examined the sensitivities of noble gas fractionations at the ground surface to the change in entire stratosphere as a first approach. We believe that the differences of changes between the shallow and deep branches of the BDC will be a future challenge, such as modeling noble gases using a 3-D model."

*Lines 511-520: It is unclear to me why the authors chose to only simulate increased mean age of air in this study, if it is unclear whether the mean mid-stratospheric age in increasing or decreasing. It would make more sense to simulate both increased and decreased mean ages, or explain more clearly why this choice was made.*

We have also conducted simulations of a decreasing mean age scenario (enhanced RMC scenario). The resulting trend is the opposite of that of the weakened RMC scenario, but the magnitude is almost the same. Similar results have been also shown in trends simulated for $\delta(Ar/N_2)$ by Ishidoya et al. (2021). Since including figures of both results in this study would be somewhat cumbersome, we have omitted the figure and added relevant descriptions in the main text. The reason we chose the weakened RMC scenario here is that, although the trend of the mean age of air in mid-stratosphere is still inconclusive, observations have reported a weak positive trend (Engel et al., 2009), which has been confirmed very recently by observations of $CO_2$-age (Sugawara et al., 2025). We have added some sentences as follows:

Lines 563-564:

", which was supported by more recent result of $CO_2$-age observation (Sugawara et al, 2025)"

Lines 581-584:

"We have also conducted simulations of enhanced-RMC scenario so that the mean age of air decreased by

0.15 years decade$^{-1}$ at an altitude of 35 km over the northern mid-latitudes. The resulting trends in $\delta_\Omega$ at the ground surface and mid-stratosphere were the opposite of those of the weakened-RMC scenario, but their magnitudes were almost the same. Similar results have been also shown in trends simulated for $\delta(Ar/N_2)$ by Ishidoya et al. (2021)."

*Line 528-530: "significant influence on estimates of ocean heat content.."*

*Is there any way to get some first order constraints on this? For e.g., would it be possible to make a statement like "if stratospheric circulation changed by X%, we would expect it to have Y% impact on estimates on long term ocean heat content" or something along these lines? This seems like a very important constraint to have on this proxy. Noble gas thermometry for mean ocean temperatures from ice cores (for e.g.: https://www.researchsquare.com/article/rs-5610580/v1) seem to fundamentally assume no other processes are affecting it. If there is indeed an effect from circulation, it would be very important to know how large this effect could be, and how that translates to effects on MOT reconstructions.*

Thank you for your professional comments on OHC. This reminded us that this point is very important, and we have added the following new section 3.6 as "Implications for ocean heat content and noble gas thermometry of mean ocean temperature". We discussed about the effects of OHC increase and BDC change on the elemental ratios of noble gases. It is only a rough estimate, but we have tried to describe it as quantitatively as possible. We have also added relevant references.
Lines 665-703:
"3.6 Implications for ocean heat content and noble gas thermometry of mean ocean temperature
Noble gases are extremely stable in the atmosphere, but they can be exchanged between the atmosphere and ocean because their solubilities in seawater change with seawater temperature variations. Because the temperature dependence of solubility is unique to each noble gas, the air-sea flux of noble gases due to changes in seawater temperature changes the elemental ratios of noble gases in the atmosphere (e.g. Keeling et al., 2004). Using this principle, the global mean ocean temperature (MOT) over the past several hundred thousand years has been reconstructed from $\delta(Kr/N_2)$, $\delta(Xe/N_2)$ and $\delta(Xe/Kr)$ of bubble air trapped in ice cores (e.g., Bereiter et al., 2018b; Shackleton et al., 2020; Haeberli et al., 2021). Ishidoya et al. (2021) reported the long-term variations of $\delta(Ar/N_2)$ observed at ground stations in 2012 – 2020 and discussed the contributions of secular changes in global ocean heat content (OHC) and BDC. They concluded that the effect of the BDC change on $\delta(Ar/N_2)$ at the ground surface cannot be ignored.
We extended the discussion of $\delta(Ar/N_2)$ by Ishidoya et al. (2021) to $\delta(Kr/Ar)$, $\delta(Xe/Ar)$, and $\delta(Ne/Ar)$, and roughly estimated the decadal-scale effects of increased OHC on the elemental ratios of noble gases, comparing them with the effects associated with BDC variations discussed in Section 3.4. The rate of

change of $N_2$ and noble gases in response to an increase in OHC depends on seawater temperature, and the relative rates of change of $N_2$, Ar, Kr, Xe, and Ne per 100 ZJ (Zeta $= 10^{21}$) of OHC are reported in Table 1 of Keeling et al. (2004). Assuming a seawater temperature of 10°C, the rates of change for $\delta(Ar/N_2)$, $\delta(Kr/Ar)$, $\delta(Xe/Ar)$, and $\delta(Ne/Ar)$ are 2.56, 6.12, 19.42, and –3.91 per meg $(100 \text{ ZJ})^{-1}$, respectively. The OHC value varies significantly depending on the depth of the ocean that is considered. Because we mainly focus on fluctuations over a relatively short timescale in this section, we used the annual OHC data reported by NOAA/NCEI up to a depth of 700 m (https://www.ncei.noaa.gov/access/global-ocean-heat-content/index.html, last access: September 18, 2025). Using this OHC data, the average rate of change in OHC over the 10-year period from 2010 to 2020 was calculated as 8.4 ZJ year$^{-1}$. Therefore, the temporal change rates of $\delta(Ar/N_2)$, $\delta(Kr/Ar)$, $\delta(Xe/Ar)$, and $\delta(Ne/Ar)$ associated with the increase in OHC during this period were estimated to be 2.2, 5.1, 16.3, and –3.3 per meg decade$^{-1}$, respectively. The rates of change in $\delta_\Omega$ caused by the change in BDC described in Section 3.4 (increase rate in Table 3) were comparable to the rates of change caused by increases in OHC. The relative magnitudes of the change rates due to the BDC change to those due to the OHC increase are 66, 71, 41, and 78 % for $\delta(Ar/N_2)$, $\delta(Kr/Ar)$, $\delta(Xe/Ar)$, and $\delta(Ne/Ar)$, respectively. This result suggests that OHC and BDC variations are essential for decadal-scale variations in the elemental ratios of noble gases.

It is interesting to determine how the BDC fluctuations can affect noble gases over the timescale of glacial-interglacial cycles. Fu et al. (2020) simulated BDC during the Last Glacial Maximum (LGM) using the Whole Atmosphere Community Climate Model (WACCM) and showed that the tropical upwelling during the LGM was weaker than that during the modern climate and that the mean age of air increased everywhere in the stratosphere during the LGM. We conducted an additional 2-D model simulation for a steady-state condition with a slow BDC in a simplified manner. The RMC in the model was weakened so that the mean age of air increased by 0.3 years (approximately 9 %) at an altitude of 35 km over the northern mid-latitudes when the model reached steady state after spin-up calculation during the 40-year period. As a result of this simulation, the changes in the annual average $\delta(Ar/N_2)$, $\delta(Kr/Ar)$, $\delta(Xe/Ar)$, and $\delta(Ne/Ar)$ at the ground surface in southern high latitudes were 2.7, 7.0, 12.8, and –4.9 per meg, respectively, compared with those before changing the RMC. If the BDC changes significantly with glacial-interglacial cycles, this may be recorded in the noble gas elemental ratios in ice core samples, which may need to be considered when the past MOT is reconstructed from noble gases."

Accounting for these discussions, we added the following to the abstract:
Lines 29-30:
"However, it was suggested that changes in the stratospheric circulation during glacial and interglacial cycles may have affected the noble gas elemental ratios in ice core samples."

---

## Author Comment (AC2)

**Author Responses to the Referee #2 Comments**

Thank you very much for your significant and useful comments on the paper "Kinetic fractionation of noble gases in the stratosphere over Japan" by Sugawara et al. We have revised the manuscript, considering your comments and suggestions. Details of our revision are as follows. The line numbers denote those of the revised manuscript.

*The paper by Sugawara et al. presents observation of noble gases in the stratosphere and discusses the effect of kinetic fractionation in explaining observed vertical gradients. The paper is well written and presents interesting data with a solid interpretation of the effects controlling the vertical gradients and in particular the effect of gravitational settling caused by kinetic fractionation of different isotopes of the noble gases. Overall, I found the paper well written, the data of high quality and the interpretation thorough. I found the section 3.3. about the two-dimensional model somewhat difficult to follow, but this may also be caused by my lack of expertise in this area. The sections 3.4. and 3.5. could be sharpened by deriving clearer conclusions, e.g. on magnitudes of changes that could be detectable. I also suggest summarizing clearer conclusions in the abstract. I recommend the paper for publication after addressing the issues mentioned here and some modifications as detailed below.*

As you pointed out, we have revised abstract and conclusions to make it clear that an impact of a changing Brewer-Dobson circulation on the noble gas elemental ratios in the modern troposphere would be hard to detect. Please see the last item of specific comments for details.

*Specific comments:*

*1. 160 and following: what is the rf gas which was used?*

Lines 166-167:
We have added a sentence:
"Here, "rf" is the reference gas which is dried natural air filled in a high-pressure cylinder (cylinder no. CRC00045) (Ishidoya and Murayama, 2014)."

*2. 172: please clarify how the difference was calculated: is this the mean absolute difference (MAD)?*

Lines 176-177:

We have revised as follows:

"The mean absolute difference of the values in 2007 was $50 \pm 24$ per meg."

3. *201: how was the CO2 mole fraction corrected for gravitational separation, how large is this effect? Please briefly explain here, even if you reference other literature.*

We have added some sentences to explain how we corrected $CO_2$ mole fraction for gravitational separation.
Lines 225-229:

"The correction of the $CO_2$ mole fraction for gravitational separation is non-negligible and can be estimated by $C \times (m - m_{air}) \times \langle \delta_G \rangle$, where C is $CO_2$ mole fraction, m and $m_{air}$ are the respective mass numbers of the molecule and air, $\langle \delta_G \rangle$ is the average gravitational separation (Ishidoya et al., 2006; Sugawara et al., 2025). The maximum depression of the $CO_2$ mole fraction due to the gravitational separation amounts to about 0.4 $\mu mol\ mol^{-1}$ at the altitudes over 30 km, assuming C = 400 $\mu mol\ mol^{-1}$ and $\langle \delta_G \rangle = -60$ per meg."

4. *245: In the case of 29N and 28N the delta-values and the normalised data values are equal. Neverthe, when comparing to normalised values, I suggest to denote the value with a subscript n.*

Line 289 and relevant descriptions, Figure 4:
All relevant descriptions of $\delta(^{29}N_2/^{28}N_2)$ in the main text and Figure 4 have been revised. Also, we added a short sentence to make it clear as follows:
"$\delta_n(^{29}N_2/^{28}N_2)\ (= \delta(^{29}N_2/^{28}N_2))$"

5. *242-257: would it make sense to summarize these values in a table?.*

Table 1 and Line 290:
As you pointed out, we have summarized $\delta_n$ values in Table 1 and add a sentence:
"The average values of $\delta_n(X/Y)$ at altitudes above 30 km are summarized in Table 1."

6. *436: I think it would be good to distinguish more clearly between molecular diffusion and eddy diffusion. In my understanding, the latter would certainly not lead to any kinetic separation and*

We have added some sentences to the beginning of this paragraph to clarify the difference between molecular and eddy diffusions:

Lines 481-484:

"To investigate the kinetic fractionation, the relative contributions of molecular and eddy diffusion are important. In an environment dominated by molecular diffusion alone, gravitational separation according to the mass number difference of molecules occurs, whereas in an environment dominated by eddy diffusion alone, separation according to molecular species is not expected to occur."

*Section 3.4.: I understand that the overall conclusion from this is that an impact of a changing Brewer-Dobson circulation on the tropospheric isotope ratio would be hard to detect. The authors have focussed on a slowing-down of the BDC, however, models consistently predict an increasing BDC. It might be worthwhile discussing the detectability of an increasing BDC.*

We completely agree with your comment. We have added some sentences to abstract and conclusions to make it clear that an impact of a changing Brewer-Dobson circulation on the tropospheric noble gases in the modern atmosphere would be hard to detect, except for $Ar/N_2$ ratio. Considering RC#1, we also performed additional simulations of the glacial condition and discussed about influences on the noble gas elemental ratios in ice core samples in Sect. 3.6. We added as follows:

Lines 28-30 (abstract):

"In the modern atmosphere, it is difficult to detect the long-term change of the stratospheric circulation from noble gases, except for $Ar/N_2$ ratio, in the troposphere. However, it was suggested that changes in the stratospheric circulation during glacial and interglacial cycles may have affected the noble gas elemental ratios in ice core samples."

Lines 741-744 (conclusions):

"At present, it is difficult to detect variations of noble gases elemental ratios in the troposphere caused by variations in stratospheric circulation, except for $\delta(^{40}Ar/^{28}N_2)$. However, when considering longer timescales such as glacial-interglacial cycles, variations in stratospheric circulation may have a significant impact on ice core data."

We have conducted additional simulations of a decreasing mean age scenario (enhanced RMC scenario). The resulting trend is the opposite of that of the weakened RMC scenario, but the magnitude is almost the same. We have added some sentences as follows:

Lines 581-584:

"We have also conducted simulations of enhanced-RMC scenario so that the mean age of air decreased by 0.15 years decade$^{-1}$ at an altitude of 35 km over the northern mid-latitudes. The resulting trends in $\delta\Omega$ at the ground surface and mid-stratosphere were the opposite of those of the weakened-RMC scenario, but their magnitudes were almost the same. Similar results have been also shown in trends simulated for $\delta(Ar/N_2)$ by Ishidoya et al. (2021)."